# Minimizing non-radiative decay in molecular aggregates through control of excitonic coupling

Yuanheng Wang [1], Jiajun Ren [2] ✉ & Zhigang Shuai [1,3] ✉

The widely known "Energy Gap Law" (EGL) predicts a monotonically exponential increase in the non-radiative decay rate ($k_{nr}$) as the energy gap narrows, which hinders the development of near-infrared (NIR) emissive molecular materials. Recently, several experiments proposed that the exciton delocalization in molecular aggregates could counteract EGL to facilitate NIR emission. In this work, the nearly exact time-dependent density matrix renormalization group (TD-DMRG) method is developed to evaluate the non-radiative decay rate for exciton-phonon coupled molecular aggregates. Systematical numerical simulations show, by increasing the excitonic coupling, $k_{nr}$ will first decrease, then reach a minimum, and finally start to increase to follow EGL, which is an overall result of two opposite effects of a smaller energy gap and a smaller effective electron-phonon coupling. This anomalous non-monotonic behavior is found robust in a number of models, including dimer, one-dimensional chain, and two-dimensional square lattice. The optimal excitonic coupling strength that gives the minimum $k_{nr}$ is about half of the monomer reorganization energy and is also influenced by system size, dimensionality, and temperature.

The suppression of non-radiative decay is a long-pursued goal in the development of near-infrared (NIR) light-emitting organic materials[1–6], because their much narrower energy gap would lead to exponentially faster non-radiative decay according to the widely known "Energy Gap Law"[7,8] (EGL). Recently, exciton delocalization as a result of excitonic couplings (*J*) between monomers in J-aggregates has been proposed as a promising pathway toward high-efficiency NIR light-emission[9,10], because this strategy not only narrows the energy gap but also reduces the effective electron-phonon coupling (*g*)/reorganization energy (*λ*), which is the dominant contributor to non-radiative decay[11,12]. This idea is supported by the experimental results by Cravcenco et al.[9], in which the reduction of reorganization energy and the suppression of non-radiative decay rate of bay-alkylated quaterrylene were observed by comparing the photophysics of monomer in toluene and J-aggregates

formed in 1,1,2,2-tetrachloroethane. Additionally, in another work by Wei et al.[10], both higher photoluminescence quantum yield (PLQY) and narrower emission energy gap in a series of NIR-emissive Pt(II) complexes have been reported. The reduction of effective reorganization energy by increasing the exciton delocalization length was proposed as the mechanism suppressing non-radiative decay and thus improving PLQY[10]. A similar suppressed non-radiative decay with increasing exciton delocalization length behavior was also observed by Patalag et al. in chain-like oligo-BODIPY superstructures[13]. In these works, the effective reorganization energy of an aggregate consisting of *N* molecules was assumed to become 1/*N* of that of a monomer. However, rigorously speaking, this simple picture is true only in the limit of strong excitonic coupling (*J* ≫ *λ*)[14]. In fact, there are few real-world molecular aggregates belonging to this strong excitonic coupling

[1]MOE Key Laboratory of Organic OptoElectronics and Molecular Engineering, Department of Chemistry, Tsinghua University, 100084 Beijing, People's Republic of China. [2]Key Laboratory of Theoretical and Computational Photochemistry, Ministry of Education, College of Chemistry, Beijing Normal University, 100875 Beijing, People's Republic of China. [3]School of Science and Engineering, The Chinese University of Hong Kong, Shenzhen 518172, People's Republic of China. ✉e-mail: jjren@bnu.edu.cn; zgshuai@tsinghua.edu.cn

regime because of the weak inter-molecular interaction. Even if this assumption is fulfilled, it is still dangerous to draw the conclusion that a large excitonic coupling/exciton delocalization length would suppress non-radiative decay because the influence of excitonic coupling on non-radiative decay has two distinct sides: (1) the excitonic coupling narrows the energy gap, which accelerates non-radiative decay; (2) the excitonic coupling decreases the effective electron-phonon coupling, which suppresses non-radiative decay. Therefore, it is important to figure out which effect is dominant in real-world molecular aggregates to minimize the non-radiative decay with rational design. As far as we know, a rigorous theoretical study is still lacking.

In the development of EGL, Englman and Jortner derived an elegant and analytical equation to approximate the non-radiative decay rate between two harmonic potential energy surfaces (PES)[7]. However, this analytical equation is unsuitable for molecular aggregates because the excitonic couplings between monomers can result in anharmonic effects on the adiabatic excited state PES and nonadiabatic effects (vibronic coupling) between excited states. In addition, since most molecular aggregates of interest are in the intermediate excitonic coupling regime (the excitonic coupling and electron-phonon coupling are comparable), obtaining a general analytical equation, which is usually derived by treating either part as a perturbation, is very difficult if not impossible. Hence, more efforts are focused on numerical solutions. However, due to the inherent electron-phonon coupled many-body problem, the affordable system size and accuracy are quite limited. The current explorations in this direction are mainly restricted to demonstrative dimer models[15,16] and the results are hard to unify. Li et al. found that non-radiative decay is enhanced by excitonic coupling in both J-aggregates and H-aggregates as predicted by EGL[15]. They combined the thermal vibrational correlation function (TVCF) method with the split-operator approximation, while the approximation is only valid when the weak excitonic coupling limit and the short time limit are fulfilled simultaneously. In another work, Celestino et al. performed simulations through the Redfield master equation and found the non-radiative decay rate would monotonically increase or decrease by excitonic effects depending on the position of non-radiative decay channel[16]. Considering the current ambiguous situation regarding this problem, it is necessary to adopt high-level methods to assess the current results and provide a more reliable and clear picture. This is the main goal of this work.

In this work, we first develop the time-dependent density matrix renormalization group (TD-DMRG)[17–20] method to evaluate the non-radiative decay rate for exciton-phonon coupled molecular aggregates. TD-DMRG has been demonstrated as a highly accurate, efficient, and robust method to simulate the quantum dynamics of electron-phonon systems at both zero temperature and finite temperature[17,20–30]. The key advantage of TD-DMRG is that the accuracy is controlled by a single parameter (the bond dimension $M$) and thus can be systematically improved to approach the exact limit[31]. In addition, the computational cost with a fixed bond dimension is polynomial with system size and thus is scalable to large systems. Through nearly-exact TD-DMRG simulations, we find that when increasing the excitonic coupling, the energy gap of molecular aggregates is reduced as expected, but the non-radiative decay rate first decreases and then levels off to reach a minimum and finally starts to increase. This anomalous non-monotonic behavior is robust across all the systems studied in this work, covering different electron-phonon couplings, vibrational frequencies, system sizes, dimensionalities, and temperatures. Therefore, we suggest that controlling the excitonic coupling strength near this minimum point is important in the development of high-efficiency NIR-emissive materials. The optimal excitonic coupling strength for this point is found around half of the monomer reorganization energy. Our work offers a concrete theoretical picture that enables the rational design of molecular aggregates with suppressed non-radiative decay resulting from exciton delocalization. It is not only

expected to facilitate the development of NIR light-emitting organic materials but also to enhance the efficiency of the light-harvesting process in organic solar cells[32] and artificial photosynthetic systems[33,34], where exciton delocalization is a common phenomenon. Additionally, this work is believed to provide basic insights for future studies on non-radiative decay of organic aggregates in optical microcavities[35,36] where the exciton-photon coupling is also a crucial factor in addition to the exciton delocalization.

## Results
### Model for aggregates
Our simulations utilize a simplified molecular aggregate model, in which each monomer is represented by a two-level electronic system coupled with two harmonic vibrational modes, unless otherwise specified. Both vibrational modes have a frequency of $\omega = 1400\ \mathrm{cm}^{-1}$, which is typical of C-C stretching vibrations and is regarded as the primary energy drain for non-radiative decay in organic dyes[37,38]. One mode has a nonadiabatic coupling $V$, which couples the two electronic states of each monomer and induces non-radiative decay. The other mode is coupled with the electronic states with electron-phonon coupling $g$. We assign each monomer an adiabatic excitation energy $E_{\mathrm{ad}} = 10\omega \approx 715\ \mathrm{nm}$, which corresponds to the energy gap between the equilibrium points of the excited state PES and the ground state PES. Most of the electronic energy will transfer to the vibrational energy of the second mode during the non-radiative decay, with larger electron-phonon coupling $g$ leading to faster non-radiative decay. Excitonic coupling between monomers could red-shift the emission spectrum of aggregates into the NIR regime, as reported in a recent work[9]. We only consider the excitonic coupling $J$ between nearest-neighbor sites, since this coupling weakens with increasing inter-molecular distance. The phase of the excitonic couplings between monomers depends on the arrangement of their transition dipole moments (TDM). For H-aggregates with sandwich-type TDM arrangement, $J$ is positive, whereas for J-aggregates with head-to-tail TDM arrangement, $J$ is negative. We examine the effects of both the magnitude and phase of excitonic couplings on the non-radiative decay of aggregates in our study. To evaluate the differences between aggregation forms, we perform simulations on the dimer, the one-dimensional (1D) chain ($N = 20$), and the two-dimensional (2D) square lattice ($N = 6 \times 6$) models with periodic boundary conditions (PBC), as depicted in Fig. 1a. Our aggregate model inherently includes the anharmonicity of aggregate excited PES as a result of excitonic coupling, as shown in Fig. 1b. To determine the non-radiative decay rate of the current models accurately, we developed the nearly exact TD-DMRG method based on our former works[20]. For more details on the Hamiltonian and numerical methods, please refer to the "Methods" section "Hamiltonian and computational algorithm".

### Molecular dimer
We start with the simplest dimer model, in which the electron-phonon coupling strengths are varied by the Huang-Rhys factor $S$ ($=g^2$), ranging from 0.5 to 2. The monomer reorganization energy $\lambda$ ($=S\omega$) of the $S = 0.5$ case is $700\ \mathrm{cm}^{-1}$, which is closest to that of the bay-alkylated quaterrylene molecule (-589 cm$^{-1}$) forming NIR-emissive J-aggregates in a recent experiment[9]. Figure 2a shows that $k_{\mathrm{nr}}^{\mathrm{Agg}}/k_{\mathrm{nr}}^{\mathrm{Mono}}$ initially decreases, then reaches a minimum, and finally increases as the relative strength of the excitonic coupling $|J|$ to the reorganization energy $\lambda$ increases, across all different electron-phonon coupling strengths. It is interesting that the ratio of the optimal excitonic coupling strength to the monomer reorganization energy $|J|_{\mathrm{OPT}}/\lambda$ giving the minimum $k_{\mathrm{nr}}^{\mathrm{Agg}}/k_{\mathrm{nr}}^{\mathrm{Mono}}$ seems insensitive to the monomer electron-phonon coupling. In all these cases it is roughly the same, around 0.6 - 0.7.

The main difference is that a larger monomer electron-phonon coupling results in a steeper slope and a larger magnitude of change in $k_{\mathrm{nr}}^{\mathrm{Agg}}/k_{\mathrm{nr}}^{\mathrm{Mono}}$ with respect to $|J|/\lambda$. Inspired by EGL for a monomer[7], the

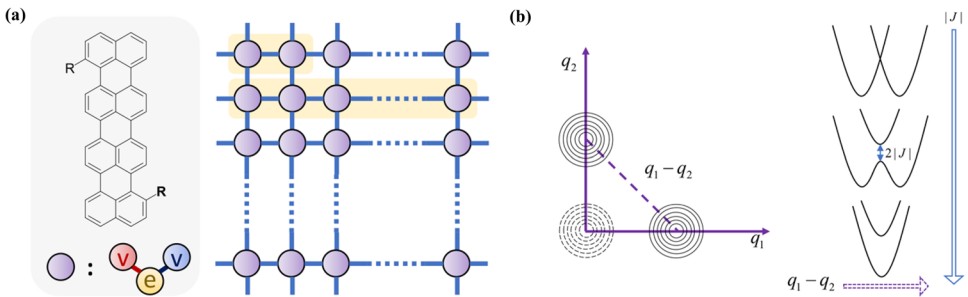

**Fig. 1 | Schematic pictures of the model for aggregates. a** A schematic picture describing the $N$-monomer aggregates studied in this work. Each monomer is approximated as a two-level electronic system (yellow circle) coupled with two harmonic vibrational modes. One of the two modes only has electron-phonon coupling $g$ and acts as the main energy drain (red circle). The other only has non-adiabatic coupling $V$ (blue circle). It couples the adiabatic electronic states and allows the non-radiative decay to happen. The periodic boundary condition (PBC) is applied for the 1D chain and 2D square lattice model. Only the nearest-neighbor excitonic coupling ($J_x = J_y = J$, $x$ and $y$ indicate the two stacking directions) is

included. **b** A schematic diagram of the potential energy surface (PES) of a dimer model. In the left panel, the dashed curves represent the ground state PES, while the solid curves represent the two excited state PESs with displacement along mode $q_1$ and $q_2$ respectively compared with the ground state. $q_1/q_2$ is the normal coordinate with electron-phonon coupling $g$ of molecule 1/ molecule 2. (the vibration represented by the red circle in (**a**)) within the dimer model. In the right panel, the 1D slice of the excited state PESs along the direction $q_1 - q_2$ with different excitonic coupling strength $|J|$ shows the anharmonicity of PES resulting from the excitonic coupling.

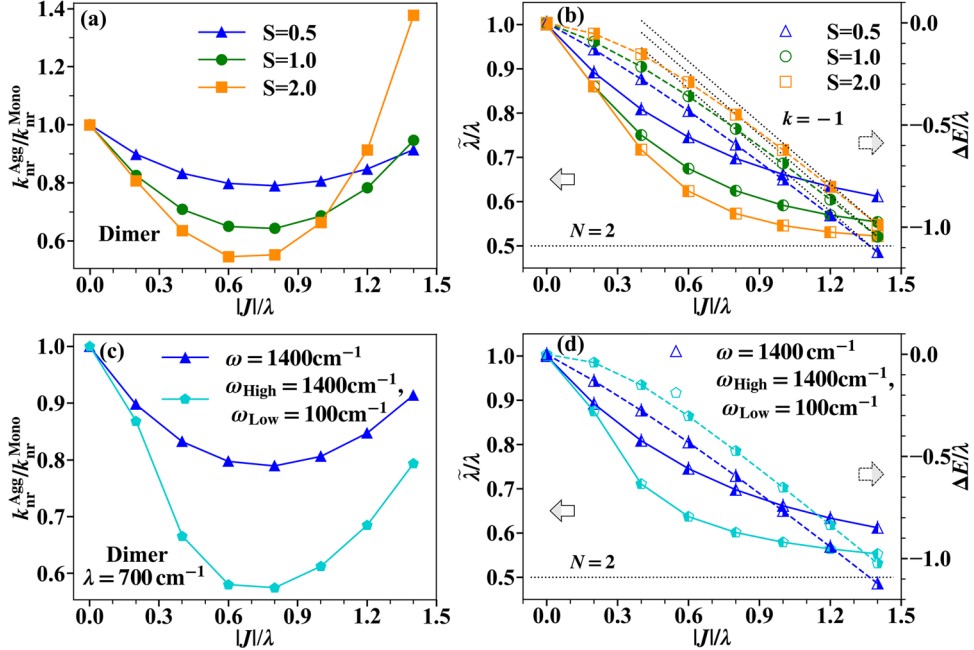

**Fig. 2 | Non-radiative decay rates for the dimer model. a** The non-radiative decay rate of dimer ($N = 2$) compared to that of monomer ($k_{nr}^{Agg}/k_{nr}^{Mono}$) in systems with different electron-phonon couplings, simulated by time-dependent density matrix renormalization group (TD-DMRG) method . ($S$ is Huang-Rhys factor quantifying the electron-phonon coupling $g$, $S = g^2$). **b** The effective aggregate reorganization energy $\tilde{\lambda}$ (solid curves, left $y$-axis) and the energy gap narrowing $\Delta E$ (dashed curves, right $y$-axis) versus the strength of excitonic coupling $|J|$. (The monomer reorganization energy $\lambda$ is used as the unit here.) The black horizontal dotted line indicates the analytical solution of $\tilde{\lambda} = \lambda/2$ in the strong excitonic coupling limit. The black dotted lines with slope $k = -1$ indicate the analytical solution of slope $k = \frac{d\Delta E}{d|J|}$ in the

strong excitonic coupling limit. The leftward arrow indicates that the left half-filled markers (solid lines) display the behavior of $\tilde{\lambda}$ using the left $y$-axis and the rightward arrow indicates that the right half-filled markers (dashed lines) display the behavior of $\Delta E$ using the right $y$-axis. **c** The blue curve corresponds to the two-mode model ($S = 0.5$) shown in (**a**) with only high-frequency modes ($\omega = \omega_{High} = 1400$ cm$^{-1}$), while the cyan curve corresponds to the three-mode model with one additional low-frequency mode ($\omega_{Low} = 100$ cm$^{-1}$). The total reorganization energies $\lambda$ of the two models are the same, and in the three-mode model, the reorganization energy is equally partitioned between the high- and low- frequency modes. **d** Same illustration as (**b**) but for systems studied in (**c**).

energy gap and reorganization energy of molecular aggregates are still expected to be the two most important factors to determine the non-radiative decay rate. Nevertheless, the reorganization energy of an aggregate is not as well defined as that of a monomer. We characterize the effective reorganization energy of an aggregate based on the vibrational distortion field (VDF) $D_n(r)$, which is the magnitude of nuclear distortion of mode $n$ at a distance $r$ from an extra electron/exciton[39-43]. VDF is well in accordance with the physical meaning of reorganization, which represents the change of equilibrium nuclear

structure when the system transitions from one electronic state to another. For the specific expression of $D_n(r)$, see "Methods" section "Vibrational distortion field". Then, the effective reorganization energy $\tilde{\lambda}$ of an aggregate is defined as $\tilde{\lambda} = \frac{1}{2}\sum_{r,n}\omega_n^2 D_n(r)^2$. In Fig. 2b, we plot the aggregate reorganization energy $\tilde{\lambda}$ and the energy gap narrowing $\Delta E = E_{0,Agg} - E_{0,Mono}$ at zero temperature ($E_{0,Agg}/E_{0,Mono}$ is the lowest energy of initial vibronic state of aggregate/monomer). Upon increasing $|J|$, $\tilde{\lambda}$ first decreases rapidly and then gradually slows down,

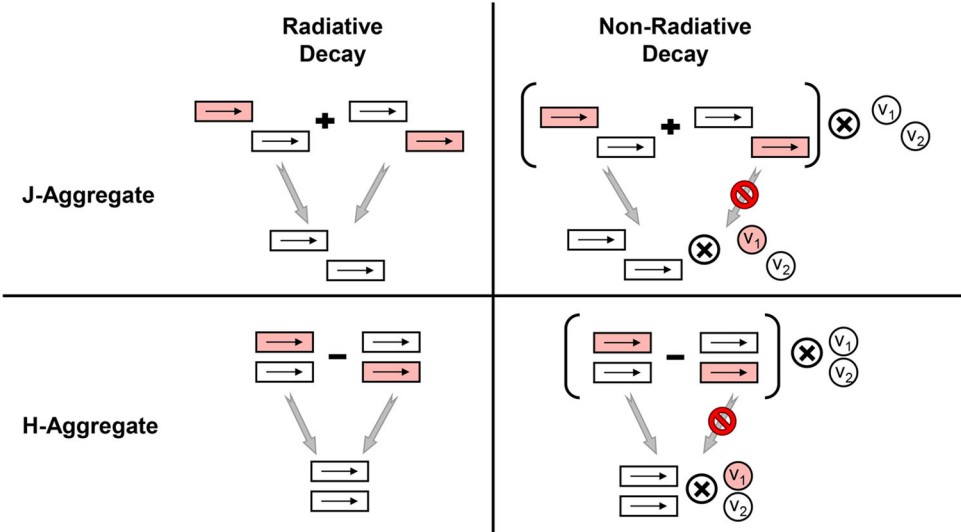

**Fig. 3 | A schematic figure illustrating the selection rule differences between the radiative decay and non-radiative decay processes of a dimer model.** Rectangles represent the monomers and the arrow inside indicates the orientation of the transition dipole moment. Each circle with $v_1$ and $v_2$ symbols represents the vibrational mode with nonadiabatic coupling for each monomer (blue circle in Fig. 1a). The direct product of electronic state (rectangles in bracket) and vibrational state (circles) forms the initial state. Because the selection rule for radiative decay is less sensitive to the vibrational state, they are not shown there. The red-colored rectangles and circles represent the local electronic and vibrational excited states, respectively. The lowest electronic excited state of dimer is a linear combination of the local excited state of monomer with different signs (±). The gray arrows indicate the coupling between the two states. The prohibition sign on the arrow means the coupling is zero.

ultimately approaching the analytical solution of $\lambda/2$ in the strong excitonic coupling limit[43]. In contrast, the energy gap narrowing $\Delta E$ is initially nearly constant but becomes steeper as $|J|$ increases. In the strong excitonic coupling limit, the slope asymptotically converges to a constant value of $k = \frac{d\Delta E}{d|J|} = -1$ [43,44]. The different varying trends of $\tilde{\lambda}/\lambda$ and $\Delta E/\lambda$ explain the non-monotonic behavior observed in Fig. 2a. As pointed out by the EGL equation ("Methods" section "Original formalism and modification of energy gap law")[7], the transition rate between two states is faster with a larger reorganization energy and a smaller energy gap. This relation, although derived for a single molecule, is expected to hold in molecular aggregates as well. Hence, in the strong excitonic coupling regime ($|J|/\lambda > |J|_{\mathrm{OPT}}/\lambda$), where the energy gap of aggregates continues decreasing with a constant slope while the effective reorganization energy $\tilde{\lambda}$ approaches a constant, the energy gap narrowing effect becomes the dominant factor. In this regime, $k_{\mathrm{nr}}^{\mathrm{Agg}}$ increases with increasing $|J|$ as predicted by the EGL equation. Conversely, in the weak excitonic coupling regime ($|J|/\lambda < |J|_{\mathrm{OPT}}/\lambda$), where $\tilde{\lambda}$ decreases rapidly with $|J|$ while $\Delta E$ decreases slowly, the reduction of effective electron-phonon coupling is the major effect. This leads to the anomalous behavior observed in $k_{\mathrm{nr}}^{\mathrm{Agg}}$, which decreases with increasing $|J|$ in this regime. (In later parts, we refer to this regime as the "anomalous regime".) Furthermore, for systems with larger $S$, the suppression of $\tilde{\lambda}$ becomes more pronounced, while the decrease in $\Delta E$ becomes slower. This explains why $k_{\mathrm{nr}}^{\mathrm{Agg}}/k_{\mathrm{nr}}^{\mathrm{Mono}}$ changes more rapidly in systems with larger $S$, as shown in Fig. 2a. The two factors also have contrary effects on the position of $|J|_{\mathrm{OPT}}/\lambda$, ultimately resulting in its overall insensitivity to $S$.

In addition to the high-frequency C-C stretching vibrations, low-frequency torsion modes with large electron-phonon couplings have also been found to play an important role in non-radiative decay for flexible molecules[45,46]. Hence, we include another low-frequency mode ($\omega_{\mathrm{Low}} = 100 \, \mathrm{cm}^{-1}$) in the former dimer model to see whether the anomalous regime still exists. For comparison, the monomer reorganization energy is equally divided into the high-frequency mode and low-frequency mode. In Fig. 2c, the results with $\lambda = 700 \, \mathrm{cm}^{-1}$

($S_{\mathrm{Low}} = 3.5$, $S_{\mathrm{High}} = 0.25$) reveal that the non-monotonic trend persists but with a greater magnitude and a steeper slope of change in $k_{\mathrm{nr}}^{\mathrm{Agg}}/k_{\mathrm{nr}}^{\mathrm{Mono}}$ as a function of $|J|/\lambda$ when the low-frequency mode is taken into account. The reason for this is attributed to the more significant suppression of $\tilde{\lambda}$, as indicated by Fig. 2d. It is consistent with the trend observed in Fig. 2a, that systems with larger Huang-Rhys factor $S$ will exhibit a greater suppression of $k_{\mathrm{nr}}$. Furthermore, it also demonstrates that non-radiative decay processes dominated by low-frequency modes are more susceptible to being suppressed by controlling exciton coupling.

Another noteworthy result is that J-aggregate ($J < 0$) and H-aggregate ($J > 0$) display the same non-radiative decay curve (see Supplementary Fig. 1 in the Supplementary Information for detailed comparisons). It is consistent with former studies[15]. This differs from the widely known enhancement/inhibition behavior in the radiative decay of J-/H- aggregates[43,44], because the nonadiabatic coupling operator, unlike the dipole operator, has both electronic and nuclear parts, resulting in a selection rule distinct from that of radiative decay, as depicted in Fig. 3. More specifically, because of excitonic coupling, the initial excited state is a linear combination of the local excited states. In the radiative decay process, one final state could have non-zero coupling with each component of the initial state (left panel in Fig. 3), therefore the sign (±) of the linear combination determines whether the emission is enhanced (+) in J-aggregate or prohibited (−) in H-aggregate. On the contrary, in the non-radiative decay process, one final state only has non-zero coupling with one component of the initial state (right panel in Fig. 3), making the sign of linear combination irrelevant. It should be noted that for non-radiative decay, the rigorous equivalence between J- and H- aggregates hold only when the modes with non-zero nonadiabatic coupling $V$ all have zero electron-phonon couplings $g$. (An analytical derivation can be found in Supplementary Note 1 in the Supplementary Information) Otherwise, J- and H- aggregates may display slightly different non-radiative decay behavior (see Supplementary Fig. 1 in the Supplementary Information). Nevertheless, in real-world organic dyes with multiple vibrational modes, the modes with strong nonadiabatic couplings often have negligible electron-phonon couplings. Thus, we believe the sign of the excitonic

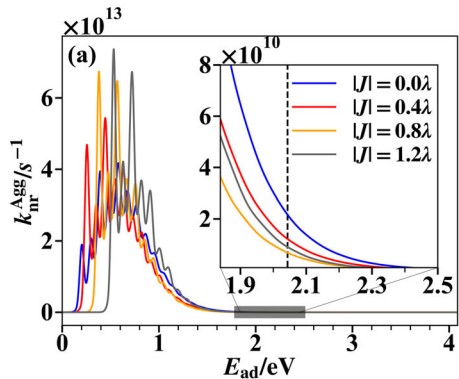
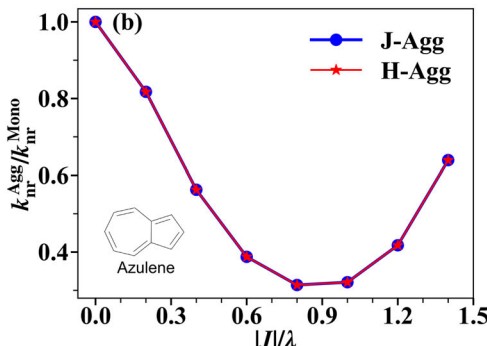

**Fig. 4 | Non-radiative decay rates for azulene dimer. a** The non-radiative decay ($k_{nr}^{Agg}$) spectrum for azulene dimers with different excitonic coupling strength $|J|$, simulated by time-dependent density matrix renormalization group (TD-DMRG) method. (The monomer reorganization energy $\lambda$ is used as the unit here.) The inset shows the enlarged window around the ab initio calculated adiabatic excitation energy $E_{ad}$ of monomer (The position is indicated by the dashed line). **b** The non-radiative decay for aggregates with different signs of excitonic coupling compared to that of monomer ($k_{nr}^{Agg}/k_{nr}^{Mono}$).

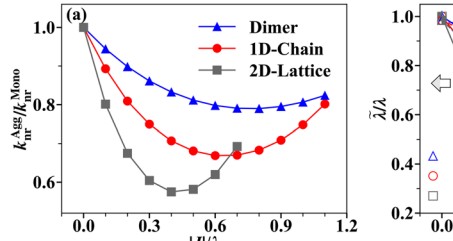
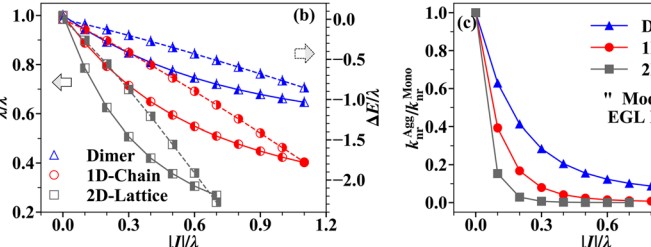

**Fig. 5 | Non-radiative decay rates for dimer, 1D chain and 2D square lattice models with different excitonic couplings strength $|J|$. a** Non-radiative decay rates of aggregates compared to that of monomer ($k_{nr}^{Agg}/k_{nr}^{Mono}$) at zero temperature for 1D chain (red curves) and 2D square lattice (gray curves) simulated by time-dependent density matrix renormalization group (TD-DMRG) method. The results of dimer (blue curves) are also shown for comparison. $S = g^2 = 0.5$ is set for electron-phonon coupling of all monomers. ($S$ is the Huang-Rhys factor and $g$ is the electron-phonon coupling) (The monomer reorganization energy $\lambda$ is used as the unit here.) **b** The effective aggregate reorganization energy $\tilde{\lambda}$ and the energy gap narrowing $\Delta E$ analysis as Fig. 2b. Also, the leftward arrow indicates the left half-filled markers display the behavior of $\tilde{\lambda}$ using the left $y$-axis and the rightward arrow indicates the right half-filled markers display the behavior of $\Delta E$ using the right $y$-axis. **c** $k_{nr}^{Agg}/k_{nr}^{Mono}$ predicted by the "modified EGL equation" using the reorganization energy and energy gap of aggregates instead of those of monomer.

coupling will have a minor effect on $k_{nr}$ of real-world molecular aggregates. This is further supported by the following azulene example shown in Fig. 4b.

To investigate whether the simplified two-mode or three-mode models are representative of real-world molecules that typically have multiple vibrational modes, we further simulate the non-radiative decay of an azulene molecular dimer with different excitonic coupling strength. The $S_1 \rightarrow S_0$ non-radiative decay of azulene monomer has been used as a benchmark system for different simulation methods in previous works[19,47,48]. Here, we build an azulene dimer model based on ab initio quantum chemistry calculations and electron-phonon coupling analysis of the azulene monomer. (See Supplementary Note 3 in the Supplementary Information for detailed ab initio calculation data.) The excitonic coupling between the two monomers is allowed to vary, ranging from 0 to 0.59 eV. It should be noted that, since the transition dipole moment between $S_1$ and $S_0$ of azulene monomer is very small, real azulene aggregates are unlikely to have such a strong excitonic coupling. We use it here only as a model system to study the effect of multiple vibrational modes. The simulated non-radiative decay spectrum is shown in Fig. 4a. The non-monotonic relation between $k_{nr}^{Agg}$ and $|J|$ still holds. As shown in the inset, when $E_{ad}$ is around the ab initio calculated value of a monomer (the dashed line), $k_{nr}^{Agg}$ decreases as $|J|$ increases from 0 to $0.8\lambda$, and then increases as $|J|$ further increases from $0.8\lambda$ to $1.2\lambda$. This behavior results from the interplay between the spectrum-narrowing effect and the blue-shift effect, both of which will be enhanced with increasing exciton coupling strength. The spectrum-narrowing effect is due to the reduction of effective electron-phonon

coupling, while the blue-shift effect is due to the decrease in the energy gap of the aggregates. In Fig. 4b, we observe that the sign of excitonic coupling has no observable effect on the non-radiative decay rate, consistent with the analysis of the two-mode model described above. This result is due to the fact that the ten modes with the largest non-adiabatic couplings among all 48 modes of azulene contribute to only 0.003% of the total reorganization energy. It is worth noting that Fig. 4b also shows that the optimal excitonic coupling strength $|J|_{OPT}$ to minimize $k_{nr}^{Agg}$ in the azulene dimer case is around $0.6\lambda \sim 0.7\lambda$, similar to those previously predicted in other dimer models in Fig. 2a, c. This suggests that the $|J|_{OPT}$ we found in simplified two-mode models is also applicable to real-world molecular aggregates. Considering the physical picture and qualitative behavior of the two-mode dimer model are well in agreement with those of the azulene dimer, we will use this two-mode model in subsequent simulations to further investigate the effect of sizes, dimensionality, and temperature.

## Effect of aggregation size and dimensionality

Next, we evaluate the relationship between $k_{nr}^{Agg}$ and $|J|$ in 1D and 2D molecular aggregates, which are more representative of real-world conditions. The periodic boundary conditions are considered in both cases. The Huang-Rhys factor $S$ is set to be 0.5 in all aggregates studied in this subsection. The results in Fig. 5a for 1D chain and 2D square lattice exhibit non-monotonic trends similar to those in the dimer model. However, the value of $|J|_{OPT}/\lambda$ for the minimum $k_{nr}^{Agg}/k_{nr}^{Mono}$ decreases as the aggregation form changes from dimer to 1D chain and from 1D chain to 2D square lattice. In addition, the slope of $k_{nr}^{Agg}/k_{nr}^{Mono}$

with respect to $|J|/\lambda$ becomes steeper and the magnitude of change becomes greater, indicating that larger system sizes and higher dimensionalities make $k_{nr}$ more sensitive to excitonic coupling. For instance, at zero temperature and when $|J|/\lambda = 0.2$, $k_{nr}^{Agg}$ is reduced to 90% of $k_{nr}^{Mono}$ in the dimer, 81% in the 1D chain, and 67% in the 2D square lattice. Figure 5b shows that $\tilde{\lambda}$ and $\Delta E$ both decrease faster with increasing $|J|$ in the 2D square lattice than in the 1D chain. The faster decrease of $\tilde{\lambda}$ means that the strong excitonic coupling regime is reached more quickly, where the value of $\tilde{\lambda}$ has converged and in turn the energy gap narrowing effect becomes dominant. This explains why $|J|_{OPT}/\lambda$ becomes smaller from the dimer to the 1D chain and then to the 2D square lattice. Nevertheless, directly predicting the difference in $k_{nr}$ between two different aggregation forms is difficult because both the two factors with opposite effects, energy gap and effective reorganization energy, both change more significantly in the 2D square lattice than in the 1D chain. Humeniuk et al. previously proposed a modified EGL equation for aggregates by using the analytical solutions of aggregate reorganization energy $\tilde{\lambda}$ and energy gap $\widetilde{E_{ad}}$ in the strong excitonic coupling limit to replace those of monomers in the EGL equation[49]. Here, we also utilize this modified EGL equation, but with $\tilde{\lambda}$ and $\widetilde{E_{ad}}$ for aggregates calculated at each excitonic coupling, attempting to qualitatively predict $k_{nr}^{Agg}/k_{nr}^{Mono}$. The definition of $\tilde{\lambda}$ was introduced earlier and $\widetilde{E_{ad}} = \Delta E + E_{ad}$. Although the EGL equation is unsuitable for aggregates due to the anharmonic and nonadiabatic effect in aggregates, we expect results from the modified equation should be qualitatively right when $|J|/\lambda$ is small, as there the electronic excited state PES is not far from the harmonic form of the monomer. Figure 5c demonstrates that when $|J|/\lambda$ is small, the modified EGL equation can capture the decreasing trend of $k_{nr}^{Agg}/k_{nr}^{Mono}$ with increasing $|J|$. Meanwhile, $k_{nr}$ of the 2D square lattice also changes the fastest, which is qualitatively consistent with our TD-DMRG numerical results. It demonstrates that in the weak excitonic coupling regime, the dominant factor influencing $k_{nr}$ is the suppression of the reorganization energy. However, the modified EGL equation severely overestimates the reduction of $k_{nr}^{Agg}$ and is unable to locate the position of $|J|_{OPT}/\lambda$ determined by TD-DMRG. Hence, we emphasize that using highly accurate methods such as TD-DMRG is necessary to describe $k_{nr}^{Agg}$ correctly in a broad parameter regime.

In Fig. 6, we plot $k_{nr}^{Agg}$ of aggregates in three aggregation forms with different $|J|$ against the energy gap $\widetilde{E_{ad}}$ of these aggregates (For convenience, the excitation energy of a monomer $E_{ad}$ is subtracted). To determine the specific locations of the optimal points (indicated by stars), we first interpolated the simulated data using the cubic spline algorithm and then identified the minimum point on the fitted curve. The curve clearly deviates from EGL, which predicts an exponentially faster $k_{nr}$ with a narrower energy gap. Instead, an anomalous regime where $k_{nr}^{Agg}$ decreases with a narrower energy gap appears when the excitonic coupling is turned on at the beginning. This result confirms the idea that properly controlling the excitonic coupling strength can lead to a narrower emission energy gap and a suppressed non-radiative decay rate in aggregates at the same time[9,10]. However, the trend eventually returns to EGL when the excitonic coupling further increases beyond an optimal value. Simulation results show that $|J|_{OPT}/\lambda$ is always smaller than 1 and is around 0.5, and that a narrower energy gap and a more suppressed $k_{nr}$ can be achieved in aggregates with higher dimensionality and larger aggregation size.

## Temperature effect

Furthermore, we simulate the non-radiative decay rate of aggregates at finite temperature to investigate the temperature effect. From the simulation results shown in Fig. 7a, we find that in all studied aggregation forms, the suppression of $k_{nr}^{Agg}$ is less significant at room temperature (RT, $k_B T = 0.15\omega = 210\,cm^{-1}$) than that at zero temperature (Fig. 5a). This behavior can be attributed to the dynamic disorder caused by electron-phonon coupling, which is enhanced at higher

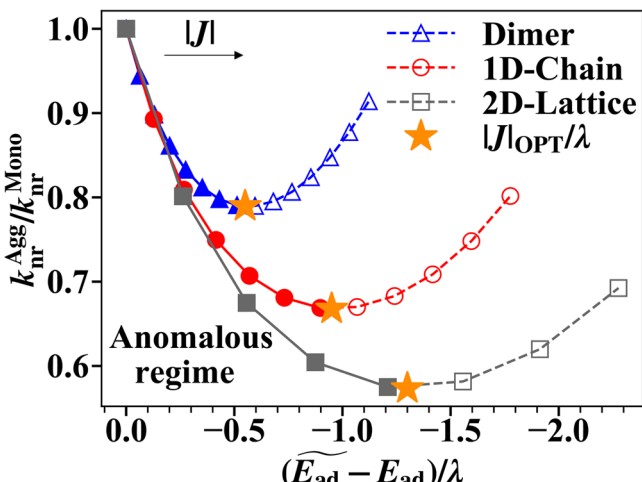

**Fig. 6 | The relation between energy gap $\widetilde{E_{ad}}$ and non-radiative decay rate $k_{nr}^{Agg}$ for different aggregate models with increasing excitonic coupling $|J|$.** The energy gap is shifted by the monomer adiabatic excitation energy $E_{ad}$. The rate is compared with the monomer non-radiative decay rate $k_{nr}^{Mono}$. The Huang-Rhys factors are all $S = g^2 = 0.5$ for each monomer in the systems. The top left arrow indicates that markers in the same curve are the results with increasing $|J|$. The difference of excitonic coupling strength between the nearest markers on the same curve is all $0.1\lambda$ and markers on the right have the stronger $|J|$ (The monomer reorganization energy $\lambda$ is used as the unit here). The anomalous regime where $k_{nr}^{Agg}$ decreases with a narrower energy gap is highlighted with solid curves and filled markers. The orange stars indicate optimal excitonic coupling strength $|J|_{OPT}/\lambda$ that minimizes $k_{nr}^{Agg}$. (For dimer, $|J|/\lambda \in \{0, 0.1, 0.2, ..., 1.4\}$ and $|J|_{OPT} \approx 0.7\lambda$. For 1D chain, $|J|/\lambda \in \{0, 0.1, 0.2, ..., 1.1\}$ and $|J|_{OPT} \approx 0.6\lambda$. For 2D square lattice, $|J|/\lambda \in \{0, 0.1, 0.2, ..., 0.7\}$ and $|J|_{OPT} \approx 0.4\lambda$).

temperatures, leading to a more localized exciton polaron (the quasi-particle describing exciton dressed by phonons)[43]. Through systematic simulations at various temperatures for the 1D chain, we observe that in the low-temperature regime where the anomalous non-monotonic behavior is evident (the lower part of Fig. 7b), the optimal excitonic coupling strength increases with temperature (the orange curve). This is because a stronger excitonic coupling is needed to counteract the enhanced dynamic disorder to achieve the same level of exciton polaron delocalization. In addition, the suppression of $k_{nr}^{Agg}/k_{nr}^{Mono}$ becomes more significant at lower temperatures. In the high-temperature limit, the quantum effect from exciton polaron delocalization that leads to the non-monotonic behavior between $k_{nr}^{Agg}$ and $|J|$ is expected to eventually disappear, and the normal EGL is restored. As a result, in the high-temperature regime (the upper part of Fig. 7b), the optimal excitonic coupling strength that minimizes $k_{nr}^{Agg}$ tends to approach zero with increasing temperature.

## Exciton delocalization size

In the preceding paragraphs, we investigate the influence of excitonic coupling, electron-phonon coupling, aggregation sizes, dimensionality, and temperature on $k_{nr}$ of aggregates. All of these factors can be reflected in one single physical quantity, namely, the delocalization size of exciton polaron. The vibrational distortion field[39–43] are calculated to characterize the exciton polaron delocalization size. VDF of the 1D chain is shown in Fig. 8a and that of the 2D square lattice is shown in Fig. 8b. In the 1D chain, even with a medium excitonic coupling strength $|J|/\lambda = 0.4$, the exciton polaron is delocalized over more than three units that have already exceeded the total size of a dimer. This shows the exciton polaron could be more delocalized in aggregates with a larger size and explains the slower $k_{nr}^{Agg}$ in 1D chain than that in a dimer with the same excitonic coupling. Qualitatively, by

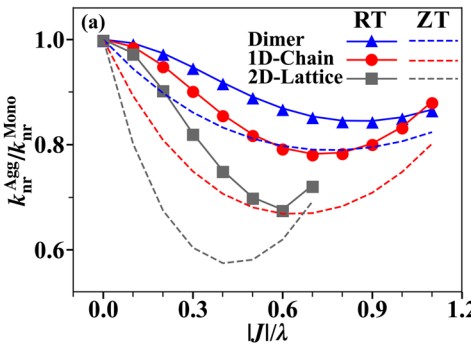
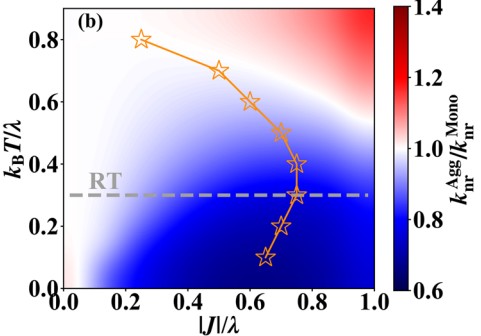

**Fig. 7 | Non-radiative decay rates for different aggregate models at finite temperature. a** The non-radiative decay rate $k_{nr}^{Agg}$ at room temperature (RT, $k_B T = 0.15\omega = 210$ cm$^{-1}$) for dimer (blue solid curves), 1D chain (red solid curves) and 2D square lattice (gray solid curves) simulated by time-dependent density matrix renormalization group (TD-DMRG) method. The results at zero temperature (ZT, dashed curves) are also shown for comparison. The Huang-Rhys factor $S = g^2 = 0.5$ is set for electron-phonon coupling of all monomers. (The monomer reorganization

energy $\lambda$ is used as the unit here to quantify the excitonic coupling strength $|J|$ and temperature $k_B T$.) **b** 2D contour showing the non-radiative decay rate $k_{nr}^{Agg}$ for the 1D chain with different excitonic coupling strength $|J|$ and at different temperatures. Orange stars are the optimal excitonic coupling at each temperature. The line connecting the stars is a guide for the eye. The gray dashed line indicates the behavior at room temperature.

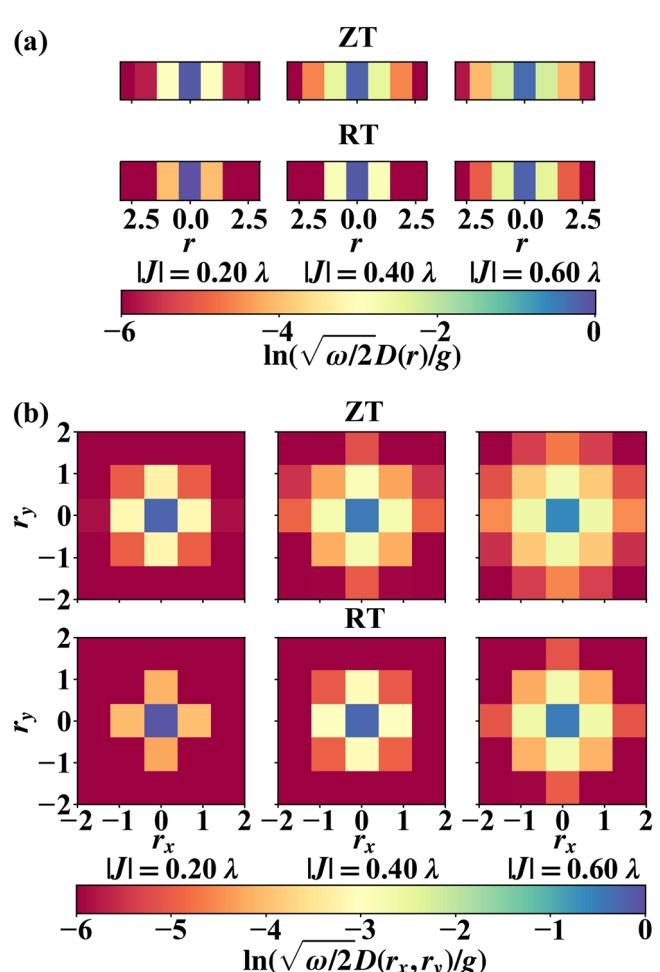

**Fig. 8 | Vibrational distortion field (VDF) $D(r)$ and $D(r_x, r_y)$ of the initial thermal equilibrium state simulated by time-dependent density matrix renormalization group (TD-DMRG) method. a** 1D chain and **b** 2D square lattice. The natural logarithm of VDF is plotted for a clear distinction of exciton polaron delocalization. (The monomer reorganization energy $\lambda$ is used as the unit here. $\omega$ is the frequency of vibration and $g$ is the electron-phonon coupling strength. ZT zero temperature, RT room temperature).

comparing Fig. 8a, b, more delocalized exciton polarons can be found in 2D square lattice than in the 1D chain. In former studies, a sum rule of VDF $\sum_r D(r) = \sqrt{\frac{2}{\omega}} g$ was analytically derived for the lowest vibronic state of 1D chain with PBC[43,50]. We find it is valid in both 1D chain and 2D square lattice by numerical calculations. Hence, $D(0)$ and $D(0, 0)$ can be used as a good descriptor to indicate the extent of delocalization of exciton polaron quantitatively. (A smaller $D(0)$ or $D(0, 0)$ means a larger delocalization size.) From Fig. 8a, b, $D(0, 0)$ in the 2D square lattice is found significantly smaller than $D(0)$ in the 1D chain at the same excitonic coupling. For instance, when $|J| = 0.4\lambda$, $D(0, 0) = 0.64g\sqrt{\frac{2}{\omega}}$ in the 2D lattice while $D(0) = 0.80g\sqrt{\frac{2}{\omega}}$ in the 1D chain. It quantitatively indicates more delocalized exciton polarons in the 2D lattice than in the 1D chain. According to the VDF distributions, the exciton polaron becomes more and more delocalized given the same $|J|$ from dimer to 1D chain and then to 2D lattice. Meanwhile, with the sum rule of VDF, the effective reorganization energy $\tilde{\lambda}$ of aggregates is smaller with a more equally distributed VDF. Therefore, a more delocalized exciton polaron is the reason for the smaller $\tilde{\lambda}$ in aggregates of higher dimensionality as shown in Fig. 5b.

**Effect of inter-molecular vibrations**

In addition to the intra-molecular vibrations that cause the fluctuation of the local excitation energy, the inter-molecular vibrations that influence the electronic coupling have also been recognized as important in affecting the delocalization of excitons or electrons[51]. Therefore, we incorporate additional low-frequency inter-molecular vibration ($\omega = 50$ cm$^{-1}$) to see whether the former physical picture will change. The Hamiltonian including inter-molecular vibrations is described in detail in "Methods" section "Hamiltonian and computational algorithm". The thermal fluctuation of $J$ is quantified by its standard deviation at a specific temperature $\Delta J = 2g_{inter}\sqrt{\omega k_B T}$[51]. We calculate $k_{nr}$ for the 1D chain with different inter-molecular electron-phonon coupling $g_{inter}$ at room temperature, making $\Delta J$ range from 0 to $0.7\lambda$. From the results in Fig. 9, we find that the inter-molecular vibrations will always enhance the effect of excitonic coupling, resulting in more suppressed $k_{nr}^{Agg}$ in the anomalous regime when $|J|/\lambda$ is small (the left part in Fig. 9) and faster $k_{nr}^{Agg}$ in the normal EGL regime when $|J|/\lambda$ is large (the right part in Fig. 9). The smaller $|J|_{OPT}/\lambda$ in the 1D chain with larger $\Delta J$ (orange curve) shown in Fig. 9 also indicates the

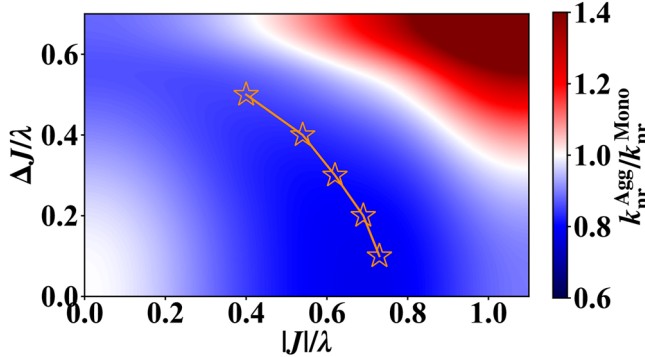

**Fig. 9 | Non-radiative decay rates $k_{nr}^{Agg}$ for the 1D chain at room temperature with different excitonic coupling $|J|$ and thermal fluctuations $\Delta J$ caused by inter-molecular vibrations.** Orange stars are the optimal excitonic coupling for the minimum non-radiative decay rate of aggregate compare to that of monomer ($k_{nr}^{Agg}/k_{nr}^{Mono}$) at different $\Delta J$. The orange line is just a guide for the eye. Results are simulated by time-dependent density matrix renormalization group (TD-DMRG) method. The monomer reorganization energy $\lambda$ is used as the unit here.

enhancement of effective excitonic coupling with a larger thermal fluctuation $\Delta J$ caused by inter-molecular vibrations.

### Connection to experiments
Finally, we connect the theoretical findings regarding the effect of excitonic coupling on the non-radiative decay rate of molecular aggregates to a recent experiment[52]. In the experiment, a series of squaraine J-aggregate homodimers (dSQA) with different bridge units are synthesized to control the distance and thus the excitonic coupling between the two monomers. The excitonic coupling strength $|J|$ and fluorescence quantum yield $\Phi_{fl}$ have been experimentally measured[52]. Here we calculated the monomer reorganization energy $\lambda$ with density functional theory and time-dependent density functional theory to compare the ratio $|J|/\lambda$ and $\Phi_{fl}$ across the series. Although the condensed phase effect and the distribution of vibration modes are neglected in the current simple analysis, we observe that the dSQA with the highest $\Phi_{fl}$ has $|J|/\lambda \approx 0.5$, which is close to the optimal exciton coupling strength suggested by our theoretical studies. See Supplementary Note 4 in the Supplementary Information for more computational details and discussions. Of course, a detailed and rigorous study of this specific system is still necessary to further unveil the connection between experiments and theories.

## Discussion
In this work, we develop the TD-DMRG method to perform nearly exact full quantum dynamics to calculate the non-radiative decay rates $k_{nr}^{Agg}$ of molecular aggregates to study its relationship with excitonic couplings $J$. In all the systems studied, including the dimer, the 1D chain, the 2D square lattice model, and the real-world molecule azulene dimer, a robust non-monotonic relationship between $k_{nr}^{Agg}$ and $|J|$ is found. In all systems studied, $k_{nr}^{Agg}$ first decreases, then levels off to reach a minimum and finally increases with increasing $|J|$. The critical optimal excitonic coupling strength $|J|_{OPT}$ to minimize $k_{nr}^{Agg}$ is less than monomer reorganization energy $\lambda$ and is estimated to be around $0.5\lambda$. Meanwhile, the emission energy gap always becomes narrower with stronger $|J|$.

From this picture, a narrow emission energy gap and largely suppressed $k_{nr}$ can be obtained simultaneously in molecular aggregates with a moderate excitonic coupling strength $|J|/\lambda \approx |J|_{OPT}/\lambda$. However, further enhancing $|J|/\lambda$ will on the contrary increase $k_{nr}$ and the behavior returns to the normal EGL, because in this regime the reduction of effective reorganization energy of aggregate $\tilde{\lambda}$ gradually approaches a constant and thus the continued narrowing energy gap becomes the dominant factor. This picture is quite different from the

former studies[9,10], in which a large excitonic coupling is believed to be essential to overcome the energy gap law. Contrary to this former understanding, our study demonstrates that the anomalous regime ($k_{nr}^{Agg}$ decreases as the energy gap decreases) will appear only in the weak to intermediate excitonic coupling regime. An intermediate excitonic coupling around the optimal point $|J|_{OPT}$ instead of strong excitonic coupling will help to suppress the non-radiative decay. This is the main finding and contribution of this work.

$|J|_{OPT}/\lambda$ is found smaller for aggregates with a larger size or higher dimensionality. The minimum value $k_{nr}^{Agg}/k_{nr}^{Mono}$ it could reach is smaller for aggregates with a larger monomer electron-phonon coupling, larger size, or higher dimensionality. The anomalous non-monotonic relationship is more obvious at lower temperatures and should be irrelevant to the sign of excitonic coupling in real-world molecular systems. Additionally, thermal fluctuation of excitonic coupling caused by inter-molecular vibrations could also make $|J|_{OPT}/\lambda$ smaller.

Our work offers a concrete theoretical picture showing that a medium strength of excitonic coupling between monomers in molecular aggregates can reduce the emission energy gap meanwhile suppress the non-radiative decay. We suggest half of the reorganization energy $\lambda$ of monomer be an optimal value of $J$ to minimize the non-radiative decay in the rational design of high-efficiency NIR-emissive organic materials, which deserves further experimental verification in the future.

## Methods
### Hamiltonian and computational algorithm
In this work, we use the Frenkel-Holstein model to describe the molecular aggregates with intra-molecular vibrations:

$$\hat{H}_0 = \sum_i E_{ad} a_i^\dagger a_i + \sum_{i,j} J a_i^\dagger a_j + \sum_{i,n} \omega_{i,n} \left( b_{i,n}^\dagger b_{i,n} + \frac{1}{2} \right) + \sum_{i,n} g_{i,n} \omega_{i,n} a_i^\dagger a_i (b_{i,n}^\dagger + b_{i,n}) + \sum_{i,n} \lambda_{i,n} a_i^\dagger a_i \quad (1)$$

where $i$ is the index of monomer. $E_{ad}$ is the adiabatic excitation energy between the ground state and the local excited state, which is considered the same for excitation on each monomer. $J$ is the excitonic coupling strength between the local excited states. We only consider the excitonic coupling between nearest-neighbor molecules. $a_i^\dagger/a_i$ corresponds to the electronic creation/annihilation operator. $n$ indicates the index of vibrational mode of each monomer $i$. $b_{i,n}^\dagger/b_{i,n}$ corresponds to the vibrational creation/annihilation operator. $\omega_{i,n}$ is its harmonic vibrational frequency, $g_{in}$ is its dimensionless electron-phonon coupling strength and $\lambda_{i,n} = g_{i,n}^2 \omega_{i,n}$ is its reorganization energy of each mode. The total reorganization energy of one monomer is $\lambda_i = \sum_n \lambda_{i,n}$. As in the monomer case, we assume that initially the system is in a thermal equilibrium state of the whole excitonic coupled Hamiltonian in Eq. (1).

When the inter-molecular vibration is included, the additional Hamiltonian term is:

$$\hat{H}_0^{inter} = \sum_i \omega_{i,inter} \left( d_i^\dagger d_i + \frac{1}{2} \right) + \sum_i g_{i,inter} \omega_{i,inter} (d_i^\dagger + d_i)(a_i^\dagger a_{i+1} + a_{i+1}^\dagger a_i) \quad (2)$$

Here we only consider one inter-molecular vibration between each nearest-neighbor sites. $\omega_{i,inter}$ and $g_{i,inter}$ is the inter-molecular vibrational frequency and electron-phonon coupling. $d_i^\dagger$ and $d_i$ are the corresponding vibrational creation/annihilation operators.

The first-order derivative term of the nonadiabatic coupling can be expressed as:

$$\hat{H}_1 = \sum_{i,n} \left( V_{i,n} \left| S_0^i \right\rangle \left\langle S_1^i \right| + h.c. \right) \hat{p}_{i,n} \quad (3)$$

$$V_{i,n} = \left\langle S_0^i | \hat{p}_{i,n} | S_1^i \right\rangle \tag{4}$$

$|S_0^i\rangle$ and $|S_1^i\rangle$ is the electronic ground and excited state of monomer $i$. Equation (3) demonstrates that the nonadiabatic coupling of molecular aggregates includes the nonadiabatic decay channel of each individual molecule. $\hat{H}_1$ is considered as the perturbation term in Fermi's Golden Rule (FGR) formalism to calculate $k_{nr}$ in this work. In the later part, we will use $|\Psi_{p/q}\rangle$ to represent the initial (one-exciton sector) and final (zero-exciton sector) vibronic state. The non-radiative decay rates $k_{nr}$ can be expressed through FGR:

$$k_{nr} = \sum_{pq} \frac{e^{-\beta E_p}}{Z} |\langle \Psi_p | \hat{H}_1 | \Psi_q \rangle|^2 \delta(E_p - E_q), \tag{5}$$

where $\beta = \frac{1}{k_B T}$ and $Z = \sum_p e^{-\beta E_p}$ is the partition function. By performing Fourier transformation on the delta functions as:

$$\delta(E_p - E_q) = \frac{1}{2\pi\hbar} \int_{-\infty}^{\infty} e^{-i(E_p - E_q)t/\hbar} dt, \tag{6}$$

we have:

$$k_{nr} = \frac{1}{\hbar^2} \int_{-\infty}^{\infty} \langle \hat{H}_1(t) \hat{H}_1 \rangle_T \, dt \tag{7}$$

$$\langle \hat{H}_1(t) \hat{H}_1 \rangle_T = \text{Tr}\left( \frac{e^{-\beta \hat{H}_0}}{Z} e^{i\hat{H}_0 t/\hbar} \hat{H}_1 e^{-i\hat{H}_0 t/\hbar} \hat{H}_1 \right) \tag{8}$$

At zero temperature, the expression can be simplified as:

$$\langle \hat{H}_1(t) \hat{H}_1 \rangle_{T=0} = e^{iE_0 t/\hbar} \left\langle 0 | \hat{H}_1 e^{-i\hat{H}_0 t/\hbar} \hat{H}_1 | 0 \right\rangle \tag{9}$$

$E_0$ and $|0\rangle$ are the lowest eigenenergy and the corresponding eigenstate of the one-exciton sector in the whole Hilbert space. The time evolution in Eqs. (8) and (9) can be solved through TD-DMRG to obtain the time correlation functions (TCF)[17–19]. At zero temperature, the initial state $|0\rangle$ and $E_0$ are solved through the DMRG ground state optimization method[31]. At finite temperature, the "purification" method[17,19,31,53] is used to represent the initial thermal equilibrium state in a matrix product form:

$$\frac{e^{-\beta \hat{H}_0}}{Z} = \frac{\text{Tr}_Q |\Psi_\beta\rangle \langle \Psi_\beta|}{\text{Tr}_{PQ} |\Psi_\beta\rangle \langle \Psi_\beta|} \tag{10}$$

$$|\Psi_\beta\rangle = e^{-\beta(H_0)_P \otimes I_Q/2} |\Psi_\infty\rangle \tag{11}$$

$$|\Psi_\infty\rangle = \prod_i \sum_{\sigma_i} |\sigma_i\rangle_P |\sigma_i\rangle_Q \tag{12}$$

$\sigma_i$ is the primitive basis for the $i$th degree of freedom. A maximally entangled state $|\Psi_\infty\rangle$ between the physical ($P$) and auxiliary ($Q$) space is used to represent the purified density matrix at infinite high temperature. Then the imaginary-time evolution is performed on $|\Psi_\infty\rangle$ to obtain the thermal state $|\Psi_\beta\rangle$ at a given temperature. Finally, the real-time evolution is performed on $|\Psi_\beta\rangle$ to obtain TCF. Please refer to our recent review for details of the methodology[20].

In this work, a phenomenological Gaussian broadening function $g(t) = e^{-\sigma^2 t^2/2}$ with $\sigma = 0.008$ a.u. (a.u. stands for atomic units) is additionally applied to the simulated TCFs of two-mode model cases and $\sigma = 0.0008$ a.u. in azulene dimer case to ensure that the TCFs decay to

zero within the simulated finite time-scale. Then we can perform time integration within this finite time window to calculate the non-radiative decay rates. Other detailed computational setups for TD-DMRG simulations are collected in Supplementary Note 2 in the Supplementary Information, including computational parameters and convergence benchmarks. The TCF before and after broadening is also shown there.

### Vibrational distortion field
The vibrational distortion field is defined as[39–43]:

$$D_n(r) = \left\langle \sum_i |S_1^i\rangle\langle S_1^i| \hat{q}_{i+r,n} \right\rangle_T. \tag{13}$$

Here, $i$ is the index of monomer, and $|S_1^i\rangle$ is the local electronic excited states of monomer $i$. $\hat{q}_{i+r,n}$ is the position operator of vibrational mode $n$ in monomer $i+r$, which has a distance $r$ from the excited monomer $i$. $\langle\rangle_T$ is the expectation value with respect to the thermal equilibrium initial state. Given the distortion $D_n(r)$, the effective reorganization energy of molecular aggregates is defined as $\tilde{\lambda} = \frac{1}{2} \sum_{r,n} \omega_n^2 D_n(r)^2$.

### Original formalism and modification of energy gap law
The EGL[7] is widely used to describe the electronic non-radiative decay rate in molecules. It is derived in the weak electron-phonon coupling limit ($g \ll 1$), low-temperature limit ($\frac{\hbar\langle\omega\rangle}{k_B T} \gg 1$), and under harmonic approximation. The rate expression for our two-mode monomer is:

$$k_{nr} = \frac{V^2 \sqrt{2\pi}}{\hbar \sqrt{\hbar\omega E_{ad}}} e^{-\frac{\lambda}{\hbar\omega}} e^{-\frac{E_{ad}}{\hbar\omega}\left[\log\left(\frac{E_{ad}}{\lambda}\right)-1\right]}. \tag{14}$$

To approximate $k_{nr}$ of aggregates, we use the effective reorganization energy $\tilde{\lambda}$ calculated from VDF and energy gap $\tilde{E}_{ad}$ of molecular aggregates to replace that of a monomer. The low-temperature limit is reasonable in studying the behavior at room temperature ($k_B T \sim 200$ cm$^{-1}$) in our systems where $\omega = 1400$ cm$^{-1}$. Though the modified equation neglects the nonadiabatic effect and the anharmonic effect that arise in molecular aggregates due to excitonic coupling, we consider the trend is at least qualitatively right with weak excitonic coupling. The limitation is discussed in the main text.

## Data availability
Source data are provided with this paper including raw data for Figs. 2, 4–9, and ab initio quantum chemistry data for molecules azulene and SQA. The data generated in this study has been deposited in Zenodo with https://doi.org/10.5281/zenodo.8042117[54].

## Code availability
The computer code for the DMRG related algorithms used in this work is available publicly via https://github.com/shuaigroup/Renormalizer[55].

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

## Acknowledgements

We thank Yu Xiong for his assistance with the manuscript presentation. Z.S. acknowledges supports from the National Natural Science Foundation of China Grant No. 21788102 and the Ministry of Science and Technology of China through the National Key R&D Plan Grant No. 2017YFA0204501. J.R. acknowledges supports from the National Natural Science Foundation of China No. 22273005.

## Author contributions

Z.S. conceived and supervised the study. Y.W. performed the numerical calculations, analyzed the data, and wrote the manuscript with the help of J.R.

## Competing interests

The authors declare no competing interests.
