## [Peer Review File · Nature Communications]

Minimizing non-radiative decay in molecular aggregates through control of excitonic couplingREVIEWER COMMENTS

Reviewer #1 (Remarks to the Author):

The authors use time dependent density matrix renormalization group (TD-DMRG) theory to describe a time correlation function (TCF). From this TCF, they derive the non-radiative decay rate. By comparing the non-radiative decay rate for different excitonic coupling strengths, and hence different energy gaps, they first, contrary to the conventional energy gap law, find a decrease with higher excitonic coupling before eventually ending up with an increase with higher excitonic coupling. They attribute this to different behaviour in two contributing factors, the reorganization energy and energy gap narrowing, with changing excitonic coupling which holds true for different aggregations and does depend on the exciton delocalization. Optimal values for the excitonic coupling are derived in the sense that they minimize the non-radiative decay rate and hence can guide sensible chemical modifications to improve near-infrared emissive molecules. The optimal values for non-radiative decay rates are in the intermediate rather than strong excitonic coupling regime.

It is interesting work that not only showcases an exciting toolbox for theoreticians but also has merits for experimentalists as it gives guidance for molecular design that fine-tunes previous results.

TD-DMRG is well suited for their approach, as the authors have also shown in previous work. The results do not contradict previous work but rather shift the focus from delocalization to the excitonic coupling itself thus complementing earlier findings. The data shown supports their conclusions. Nonetheless, I feel like some points should be expanded upon.

1. While the non-radiative decay rate is obtained from a TCF, the TCF itself is never shown. I would appreciate some select plots of the TCF before and after broadening, e.g., in sec. 2.5 of the SI.
2. The authors mention the variance of $k_{nr}^{agg}/k_{nr}^{mono}$ and more importantly the trends of variance in the main text but it is not shown in the figures. While not crucial for the interpretation, it would still be nice to see it.
3. An important aspect in their analysis is the different behaviour of the renormalization energy and the energy gap narrowing. They clearly show this in Fig. 2b for the dimer and in Fig. 2d for the 2D lattice. For the 1D chain it is not as obvious and Fig. 2d might benefit from the remaining three data points of the 1D chain (cf. panel c) that are presently not shown.

Another point is about one of their key findings: the optimal value of the excitonic coupling. While they show the optimal values in Fig. 3 and mention them in the text, it is not quite clear to me how they are actually obtained. Is it the minimum of a fit to the non-radiative decay curves, just the lowest data point in the curves or something more complex? It would be much appreciated if the authors could make this clearer.

An analysis of the temperature dependence of the optimal excitonic coupling value would also be appreciated, seeing that they already have the necessary decay rate curves at room temperature.

I am a bit puzzled about the authors' choice of bond dimension for the 2D lattice following Fig. S8. I would argue that from the figure they should start using the more accurate calculations starting from $|J|/\lambda=0.5$ and not 0.6. I do not believe that this will impact the results though.

There are a few other suggestions I would like to make to hopefully improve the paper: The authors write that in the high-temperature limit, the quantum effects and anomalous behaviour should disappear. It would be interesting to have a comparative higher temperature calculation in the paper to check on this.

In my opinion, figure 3 would benefit from having the energy gap as abscissa instead of the energy gap narrowing. I understand that this is just a constant shift and could be left out. But at first glance, the current choice could lead to confusion as the magnitude of ΔE (which could be the energy gap itself in other contexts) decreases from left to right, while the actual energy gap decreases.

I would appreciate if the authors would state in the SI which excited state of azulene was used (in addition to the main text) and briefly comment on the choice of functional and basis set.

Reviewer #2 (Remarks to the Author):

The manuscript titled "Minimizing non-radiative decay in molecular aggregates through optimal control of excitonic coupling" concerns the very interesting and important problem of developing efficient NIR fluorophores. Some recent studies have shown a great potential for J-aggregates in this field due to the exciton nature of their optical properties and red-shift of the excitonic band (J-band). Also, it was demonstrated that the exciton delocalization length can be a very important characteristic that can assist to overcome the problem of quantum yield decreasing due to the well-known energy gap law.

The current manuscript proposes a model describing the competing contributions of exciton delocalization and energy gap law to non-radiative decay.

The results, obtained in the manuscript, are interesting and undoubtedly have significant importance to molecular and chemical physics. However, the manuscript, in my opinion, is related rather to the specific scientific field than to a broad audience, supposed for Nature Communication journal.

Reviewer #3 (Remarks to the Author):

Near-infrared (NIR) emissive molecular materials are desirable for technologies, however, increased non-radiative decay rates with lowering gaps is a negative factor that needs to be understood and overcome. The authors offer a modeling study using high fidelity TD-DMRG approach to explain the relationship of non-radiative decay rates with excitonic couplings in molecular aggregates, where the effective small electron-phonon coupling suppresses the non-radiative relaxation. Such observations add valuable theoretical insights into decades-long problem and may have consequences for future organic lighting applications in NIR region and perhaps other applications. Overall, the computational results are convincing and can serve as guide to experiments for designing new materials. Consequently, the article potentially may be published in Nature Communications.

However, the results are not well presented, and the manuscript is poorly written. As such, I do not recommend this paper for publication in Nature Communications in its present form. I hope that the authors will be able to address these issues in the revision.

Critique 1: While the theoretical approach (TD-DMRG) is rigorous, the effects of vibronic non-radiative relaxation are described with essentially 2 high-frequency modes the accepting (AM) and promoting (PM) mode. As the authors argue many times, the molecular aggregate system is quite complex and there are different classes of vibrational motions can contribute to the relaxation. In addition to the bond-stretching motion, like C-C stretches the authors incorporate, there are many other vibrational motions contributing to electron-phonon dynamics, such as torsional and intermolecular motions. The discussion of the physical picture and validity of the present approximation is missing. Further, anharmonicities are mentioned but are not discussed in the context of the present model.

For example, by looking at the molecule shown in Fig. 1 (fused aromatic rings, essentially hydrocarbon system), I would say that the present model should work, since there would be no torsional motions and the absence of hetero atoms would reduce impact of intermolecular

motions/electrostatic interactions. Subsequently, the reader would immediately raise the question how general the model is and if the present approach can be extended to other molecular families.

I suggest to thoroughly revise the beginning of the section Results. Instead of starting from the Hamiltonian model description (that immediately avalanches specific jargon: Frenkel-Holstein, Condon's approximation, AM/PM, Fermi's Golden Rule, Huang-Rhys factors – by the way, the displacements S are not even introduced, etc.), the authors should provide an accessible discussion for a reader of the underpinning physical picture and outline advantages and limitations of the presented model.

Well, after reading further, the discussion about multiple vibrational modes does appear much later (page 13) and is partially addressed on the azulene example (again fused rings hydrocarbons), however, such arrangement negatively affects the readability of the paper.

Critique 2:

By continuing reading, I see

1. Eq. 1. H_1 is not introduced. One must read through the Method section to understand its meaning.
2. Eq (2) - the physical meaning of reorganization – is not explained at all. Vibrational creation/annihilation operators b are only introduced at the end, in the Methods section.
3. Eq. (3) - vibrational frequencies are not introduced (the equation cannot be understood without Methods section).

Altogether the above 3 comments are suggestive that the authors started with quite technical manuscript and spitted a formal part into "Results" and "Methods", which adversely impacted readability since one needs to jump from one place to another to understand the narrative.

To contrast, separation of the Main text and Methods should serve an opposite purpose: improved readability. The Main text should be self-contained (!) and written in accessible language, opening the paper to a broader audience such as experimentalists. In contrast the Methods should provide 'in depth' look to theory practitioners.

Bottom line: the text in Results section must be appropriately revised to address the above problems.

Critique 3:

Methods section has highly technical subsection B on the "Influence of the sign of excitonic coupling", which is highly relevant to H- and J-aggregation. However, discussion of H- and J-aggregate cases is largely missing from the Main text. This is an extremely important topic that needs to be addressed in accessible terms.

Critique 4:

Toward the end of the Discussion section, it would be nice to connect formal theoretical findings on J_{OPT}/λ , etc. with realistic molecular aggregate systems for which most of the discussed quantities are known.

Response to Reviewers' Comments

Reviewer 1

The authors use time dependent density matrix renormalization group (TD-DMRG) theory to describe a time correlation function (TCF). From this TCF, they derive the non-radiative decay rate. By comparing the non-radiative decay rate for different excitonic coupling strengths, and hence different energy gaps, they first, contrary to the conventional energy gap law, find a decrease with higher excitonic coupling before eventually ending up with an increase with higher excitonic coupling. They attribute this to different behaviour in two contributing factors, the reorganization energy and energy gap narrowing, with changing excitonic coupling which holds true for different aggregations and does depend on the exciton delocalization. Optimal values for the excitonic coupling are derived in the sense that they minimize the non-radiative decay rate and hence can guide sensible chemical modifications to improve near-infrared emissive molecules. The optimal values for non-radiative decay rates are in the intermediate rather than strong excitonic coupling regime.

It is interesting work that not only showcases an exciting toolbox for theoreticians but also has merits for experimentalists as it gives guidance for molecular design that fine-tunes previous results. TD-DMRG is well suited for their approach, as the authors have also shown in previous work. The results do not contradict previous work but rather shift the focus from delocalization to the excitonic coupling itself thus complementing earlier findings. The data shown supports their conclusions. Nonetheless, I feel like some points should be expanded upon.

-1. While the non-radiative decay rate is obtained from a TCF, the TCF itself is never shown. I would appreciate some select plots of the TCF before and after broadening, e.g., in sec. 2.5 of the SI.

Our response:

We thank the Reviewer for the suggestion. In Figure S2, we now demonstrate the typical time evolution for the correlation functions of non-radiative decay for azulene dimer for different excitonic coupling strengths $|J|$ at zero temperature, before and after applying a Gaussian form broadening function.

Figure S2: The time correlation functions for non-radiative decay of azulene dimer with different excitonic coupling strength $|J|$ at zero temperature before (bottom panel) and after (upper panel) applying a gaussian form broadening function.

-2. The authors mention the variance of $k_{nr}^{agg}/k_{nr}^{mono}$ and more importantly the trends of variance in the main text but it is not shown in the figures. While not crucial for the interpretation,

it would still be nice to see it.

Our response:

We apologize for the misleading expression of “variance” in describing the nonradiative decay rate change from monomer to aggregate. In deed, the vertical axis for Fig. 2a is a ratio ($k_{nr}^{Agg}/k_{nr}^{Mono}$) which depicts the effect of excitonic coupling on the non-radiative decay in aggregate. Now, we remove the confusing wording of “variance” and use “ratio” wherever appropriate. It is the trends of this ratio that is important.

-3. An important aspect in their analysis is the different behaviour of the renormalization energy and the energy gap narrowing. They clearly show this in Fig. 2b for the dimer and in Fig. 2d for the 2D lattice. For the 1D chain it is not as obvious and Fig. 2d might benefit from the remaining three data points of the 1D chain (cf. panel c) that are presently not shown.

Our response:

Thank you for your kindly advise. We have added more data points for the 1D chain in the figure (new Figure 5b in the revised manuscript) to more clearly demonstrate the varying trend of reorganization energy and energy gap narrowing with the excitonic coupling.

Figure 5: Non-radiative decay rates for dimer, 1D chain and 2D square lattice model with different $|J|$.

a) k_{nr}^{Agg} at zero temperature simulated by TD-DMRG for 1D chain (red curves) and 2D square lattice (black curves). The results of dimer (blue curves) are also shown for comparison. $S = g^2 = 0.5$ is set for electron-phonon coupling of all monomers. b) $\tilde{\lambda}$ and ΔE analysis as FIG.2b. c) $k_{nr}^{Agg}/k_{nr}^{Mono}$ predicted by the “modified EGL equation” using the reorganization energy and energy gap of aggregates instead of those of monomer.

-4. Another point is about one of their key findings: the optimal value of the excitonic coupling. While they show the optimal values in Fig. 3 and mention them in the text, it is not quite clear to me how they are actually obtained. Is it the minimum of a fit to the non-radiative decay curves, just the lowest data point in the curves or something more complex? It would be much appreciated if the authors could make this clearer

Our response:

Thank you for the comment. We have added an explanation in the revised manuscript. “To

determine the specific locations of the optimal points (indicated by stars), we first interpolated the simulated data using the cubic spline algorithm and then identified the minimum point on the fitted curve.”

-5. An analysis of the temperature dependence of the optimal excitonic coupling value would also be appreciated, seeing that they already have the necessary decay rate curves at room temperature.

Our response:

Thank you for the insightful suggestions. We carry out additional systematic simulations at various temperatures for the 1D chain to discuss the temperature dependence of the optimal excitonic coupling value in the revised manuscript. The results are collected in Figure 7b. Orange stars are the optimal excitonic coupling at each temperature. We observe that in the low-temperature regime where the anomalous non-monotonic behavior is evident (the lower part of Figure 7b), the optimal excitonic coupling strength increases with temperature (the orange curve). This is because a stronger excitonic coupling is needed to counteract the enhanced dynamic disorder to achieve the same level of exciton polaron delocalization. In addition, the suppression of $k_{nr}^{Agg}/k_{nr}^{Mono}$ becomes more significant at lower temperatures. In the high-temperature limit, the quantum effect from exciton polaron delocalization that leads to the non-monotonic behavior between k_{nr}^{Agg} and $|J|$ is expected to eventually disappear, and the normal EGL is restored. As a result, in the high-temperature regime (the upper part of Figure 7b), the optimal excitonic coupling strength that minimizes k_{nr}^{Agg} tends to approach zero with increasing temperature.

We have added a new subsection **Temperature Effect** in the revised manuscript.

Figure 7: Non-radiative decay rates for different aggregate models at finite temperature.

a) k_{nr}^{Agg} at room temperature (RT, $k_B T = 0.15\omega = 210\text{cm}^{-1}$) simulated by TD-DMRG for dimer (blue solid curves), 1D chain (red solid curves) and 2D square lattice (black solid curves). The results at zero temperature (ZT, dashed curves) are also shown for comparison. $S = g^2 = 0.5$ is set for electron-phonon coupling of all monomers. b) 2D contour showing k_{nr}^{Agg} for the 1D chain with different excitonic coupling strength $|J|$ and at different temperatures. Orange stars are the optimal excitonic coupling at each temperature. The line connecting the stars is a guide for the eye. The grey dashed line indicates the behavior at room temperature.

-6. I am a bit puzzled about the authors' choice of bond dimension for the 2D lattice following Fig. S8. I would argue that from the figure they should start using the more accurate calculations starting from $|J|/\lambda=0.5$ and not 0.6. I do not believe that this will impact the results though.

Our response:

Thank you for your question. We carry out more accurate calculations with $M=80$ for this 2D-lattice to further check the convergence of the computational parameters of TD-DMRG. The benchmark results are shown in Figure S9 in the revised manuscript. From the benchmark result, we choose to use $M=50$ for $|J|/\lambda < 0.4$ cases, $M=70$ for $|J|/\lambda = 0.4 \sim 0.6$ cases, and more accurate $M=80$ for $|J|/\lambda = 0.7$ case to plot the corresponding Figure 7a in the revised manuscript.

Figure S9: Benchmark $k_{nr}^{Agg}/k_{nr}^{Mono}$ results for different bond dimension M used in TD-DMRG simulation in 2D square lattice at room temperature. The $M = 50$ result is qualitatively right compared to that of larger M when $|J|/\lambda < 0.4$. So we use $M = 50$ for $|J|/\lambda < 0.4$ and a more accurate result from $M = 70$ for $|J|/\lambda = 0.4 \sim 0.6$ and $M = 80$ for $|J|/\lambda = 0.7$ in the main text.

-7. There are a few other suggestions I would like to make to hopefully improve the paper:

The authors write that in the high-temperature limit, the quantum effects and anomalous behaviour should disappear. It would be interesting to have a comparative higher temperature calculation in the paper to check on this.

Our response:

Thank you for the insightful suggestion. We have added higher temperature calculations to check on this point. From the results collected in Figure 7b, anomalous non-monotonic behavior is absent at high temperature ($k_B T > 0.8\lambda$). The revised figures and related discussions are merged into our response to comment 5.

-8. In my opinion, figure 3 would benefit from having the energy gap as abscissa instead of the energy gap narrowing. I understand that this is just a constant shift and could be left out. But at first glance, the current choice could lead to confusion as the magnitude of ΔE (which could be the energy gap itself in other contexts) decreases from left to right, while the actual energy gap decreases.

Our response:

Thank you for this suggestion. In Figure 6 of the revised manuscript, we avoid using the misleading symbol ΔE as the x-axis. Instead, we use the energy gap \widetilde{E}_{ad} of the aggregates as the x-axis. However, for convenience, the excitation energy of a monomer E_{ad} is still subtracted from \widetilde{E}_{ad} , to highlight the effect of electronic coupling that leads to a reduction in the energy gap of the aggregate relative to that of the monomer.

Figure 6: The relation between energy gap and non-radiative decay rate for different aggregate models when increasing excitonic coupling $|J|$.

The Huang-Rhys factors are all $S = g^2 = 0.5$ for each monomer in the systems. The top left arrow indicates that markers in the same curve are the results with increasing $|J|$. The difference of excitonic coupling strength between the nearest markers on the same curve is all 0.1λ and markers on the right have the stronger $|J|$. The anomalous regime where k_{nr}^{Agg} decreases with a narrower energy gap is highlighted with solid curves and filled markers. The orange stars indicate optimal excitonic coupling strength $|J|_{OPT}/\lambda$ that minimizes k_{nr}^{Agg} . (For dimer, $|J|/\lambda \in \{0, 0.1, 0.2, \dots, 1.4\}$ and $|J|_{OPT} \approx 0.7\lambda$. For 1D chain, $|J|/\lambda \in \{0, 0.1, 0.2, \dots, 1.1\}$ and $|J|_{OPT} \approx 0.6\lambda$. For 2D square lattice, $|J|/\lambda \in \{0, 0.1, 0.2, \dots, 0.7\}$ and $|J|_{OPT} \approx 0.4\lambda$.)

-9. I would appreciate if the authors would state in the SI which excited state of azulene was used (in addition to the main text) and briefly comment on the choice of functional and basis set.

Our response:

Thank you for the comment. In section 3 of the revised supporting information, we add the following phrase “*We evaluated the non-radiative decay from excited state S_1 to ground state S_0* ”. We also add a short discussion on the adiabatic excitation energy of azulene calculated by different functional (B3LYP, M062X, ω B87XD) and basis sets (6-31G(d), 6-311+G(d,p)). We found that the adiabatic excitation energies are only slightly different with different functionals and basis sets, and thus we believe B3LYP/6-31G(d) is sufficiently accurate to describe the excited states of azulene.

Reviewer: 2

The manuscript titled "Minimizing non-radiative decay in molecular aggregates through optimal control of excitonic coupling" concerns the very interesting and important problem of developing efficient NIR fluorophores. Some recent studies have shown a great potential for J-aggregates in this field due to the exciton nature of their optical properties and red-shift of the excitonic band (J-band). Also, it was demonstrated that the exciton delocalization length can be a very important characteristic that can assist to overcome the problem of quantum yield decreasing due to the well-known energy gap law.

The current manuscript proposes a model describing the competing contributions of exciton delocalization and energy gap law to non-radiative decay.

The results, obtained in the manuscript, are interesting and undoubtedly have significant importance to molecular and chemical physics. However, the manuscript, in my opinion, is related rather to the specific scientific field than to a broad audience, supposed for Nature Communication journal.

Our response:

Thank you for your comments on our manuscript. We appreciate your recognition of the importance of the problem we have addressed and the potential impact of our work on molecular and chemical physics. We acknowledge that the previously submitted manuscript may include too many technical details, but we believe it could still be of interest to a wider audience, particularly those interested in the development of efficient Near-Infrared (NIR) fluorophores. We have since revised the manuscript to better cater to a broader readership while still maintaining the technical accuracy and rigor of the content. Now the main text is self-contained, includes more discussion on the extensibility of our results to different circumstances and molecular systems, and only consists few symbols/concepts like Huang-Rhys factors, reorganization energy, and energy gap that have been already widely used in related theoretical and experimental community. The technique parts are largely moved to the Method section to provide in-depth information for theory practitioners. We will continue to consider your feedback as we further refine the manuscript for publication. Thank you again for your valuable input.

Reviewer: 3

Near-infrared (NIR) emissive molecular materials are desirable for technologies, however, increased non-radiative decay rates with lowering gaps is a negative factor that needs to be understood and overcome. The authors offer a modeling study using high fidelity TD-DMRG approach to explain the relationship of non-radiative decay rates with excitonic couplings in molecular aggregates, where the effective small electron-phonon coupling suppresses the non-radiative relaxation. Such observations add valuable theoretical insights into decades-long problem and may have consequences for future organic lighting applications in NIR region and perhaps other applications. Overall, the computational results are convincing and can serve as guide to experiments for designing new materials. Consequently, the article potentially may be published in Nature Communications

However, the results are not well presented, and the manuscript is poorly written. As such, I do not recommend this paper for publication in Nature Communications in its present form. I hope that the authors will be able to address these issues in the revision.

-Critique 1: While the theoretical approach (TD-DMRG) is rigorous, the effects of vibronic non-radiative relaxation are described with essentially 2 high-frequency modes the accepting (AM) and promoting (PM) mode. As the authors argue many times, the molecular aggregate system is quite complex and there are different classes of vibrational motions can contribute to the relaxation. In addition to the bond-stretching motion, like C-C stretches the authors incorporate, there are many other vibrational motions contributing to electron-phonon dynamics, such as torsional and intermolecular motions. The discussion of the physical picture and validity of the present approximation is missing. Further, anharmonicities are mentioned but are not discussed in the context of the present model.

For example, by looking at the molecule shown in Fig. 1 (fused aromatic rings, essentially hydrocarbon system), I would say that the present model should work, since there would be no torsional motions and the absence of hetero atoms would reduce impact of intermolecular motions/electrostatic interactions. Subsequently, the reader would immediately raise the question how general the model is and if the present approach can be extended to other molecular families.

Our response:

Thank you for your valuable feedback. We have made significant revisions to the manuscript to address your suggestions. Specifically, we have added more simulations and detailed discussions about the validity of the present approximation, which involves simplifying each monomer in the aggregate as a two-level system coupled with two high-frequency modes. In the revised manuscript, we have also included the influence of low-frequency torsion modes and inter-molecular motions in our simulations. These revisions aim to provide a more comprehensive understanding of the theoretical model we used and its applicability to various molecular systems. We hope that these additions will enhance the scientific rigor and readability of the manuscript.

1. In addition to the high-frequency C-C stretching vibrations, low-frequency torsion modes with large electron-phonon couplings have also been found to play an important role in non-radiative decay for flexible molecules. [Hong, et al, Chem. Soc. Rev. **40**, 5361-5388 (2011); Deng, et al, J. Chem. Phys. **135**, 014304 (2011)]. Hence, we include another low-frequency mode ($\omega_{Low} =$

100 cm^{-1}) in the former dimer model to see whether the anomalous regime still exists. In Figure 2c, the results with $\lambda = 700 \text{ cm}^{-1}$ are shown ($S_{Low} = 3.5$ $S_{High} = 0.25$), reveal that the non-monotonic trend persists but with a greater magnitude and a steeper slope of change in $k_{nr}^{Agg}/k_{nr}^{Mono}$ as a function of $|J|/\lambda$ when the low-frequency mode is taken into account. From the effective reorganization energy $\tilde{\lambda}$ and energy gap narrowing ΔE shown in Figure 2d, the reason for this is attributed to the more significant suppression of $\tilde{\lambda}$. It is consistent with the trend observed in Figure 2a, that systems with larger Huang-Rhys factor S will exhibit a greater suppression of k_{nr}^{Agg} . Furthermore, it also demonstrates that non-radiative decay process dominated by low-frequency modes are more susceptible to being suppressed by controlling exciton coupling.

Figure 2: Non-radiative decay rates for dimer model.

a) The non-radiative decay rate of dimer ($N = 2$) compared to that of monomer ($k_{nr}^{Agg}/k_{nr}^{Mono}$) simulated by TD-DMRG in systems with different electron-phonon couplings. b) The effective aggregate reorganization energy $\tilde{\lambda}$ (solid curves, left y-axis) and the energy gap narrowing ΔE (dashed curves, right y-axis) versus the strength of excitonic coupling $|J|/\lambda$. The black horizontal dotted line indicates the analytical solution of $\tilde{\lambda} = \lambda/2$ in the strong excitonic coupling limit. The black dotted lines with slope $k = -1$ indicate the analytical solution of slope $k = \frac{d\Delta E}{d|J|}$ in the strong excitonic coupling limit. c) The blue curve corresponds to the two-mode model ($S = 0.5$) shown in (a) with only high-frequency modes, while the cyan curve corresponds to the three-mode model with one additional low-frequency mode. The total reorganization energies λ of the two models are the same, and in the three-mode model, the reorganization energy is equally partitioned between the high- and low- frequency modes. d) same as (b) but for systems studied in (c).

2. In addition to the intra-molecular vibrations that cause the fluctuation of the local excitation energy, the inter-molecular vibrations that influence the electronic coupling have also been recognized as important in affecting the delocalization of excitons or electrons. [Arag3, et al, Adv. Funct. Mater. 26, 2316-2325 (2016)] Therefore, we incorporate additional low-frequency inter-molecular vibration ($\omega = 50 \text{ cm}^{-1}$) to see whether the former physical picture will change in the revised manuscript. The Hamiltonian including inter-molecular vibrations is

described in detail in revised Methods section. The thermal fluctuation of J is quantified by its standard deviation at a specific temperature $\Delta J = 2g_{inter}\sqrt{\omega k_B T}$. [Aragó, et al, Adv. Funct. Mater. 26, 2316-2325 (2016)]. We calculate k_{nr}^{Agg} for the 1D chain with different electron-phonon coupling g_{inter} at room temperature, making ΔJ range from 0 to 0.7λ . From the results in Figure 9, we find that the inter-molecular vibrations will always enhance the effect of excitonic coupling, resulting in more suppressed k_{nr}^{Agg} in the anomalous regime when $|J|/\lambda$ is small (the left part in Figure 9) and faster k_{nr}^{Agg} in the normal EGL regime when $|J|/\lambda$ is large (the right part in Figure 9). The smaller $|J|_{OPT}/\lambda$ in the 1D chain with larger ΔJ (orange curve) shown in Figure 9 also indicates the enhancement of effective excitonic coupling with a larger thermal fluctuation ΔJ caused by inter-molecular vibrations.

Figure 9: k_{nr}^{Agg} for the 1D chain at room temperature with different excitonic coupling $|J|$ and thermal fluctuations ΔJ caused by inter-molecular vibrations.

Orange stars are the optimal excitonic coupling for the minimum $k_{nr}^{Agg} / k_{nr}^{Mono}$ at different ΔJ with line as a guide for the eye.

- Regarding the anharmonicity aspect, the Frenkel-Holstein Hamiltonian utilized in our study inherently incorporates the anharmonicity of the adiabatic excited state potential energy surface (PES) of the molecular aggregate when the excitonic coupling between the local excited states is turned on. To illustrate this, we have included a schematic diagram in Figure 1b of the revised manuscript. The inclusion of anharmonicity is one of the reasons why methods based on the harmonic approximation of the PES, such as Energy Gap Law equations, is unsuitable for molecular aggregates.

Figure 1: Schematic pictures of the model for aggregates used for simulation.

a) A schematic picture describing the N -monomer aggregates studied in this work. Each molecular monomer is approximated as a two-level electronic system (yellow circle) coupled with two harmonic vibrational modes. One of the two modes has only electron-phonon coupling g and acts as main energy drain. The other has only nonadiabatic coupling V . It couples the adiabatic electronic states and allow the non-radiative decay to happen. The periodic boundary condition (PBC) is applied for the 1D chain and 2D square lattice model. Only the nearest-neighbor excitonic coupling ($J_x = J_y = J$) is included. b) A schematic diagram of the PES of the dimer model. In the left panel, the dashed curves represent the ground state PES, while the solid curves represent the two excited state PESs with displacement along mode q_1 and q_2 respectively. In the right panel, the 1D slice of the excited state PESs along the direction $q_1 - q_2$ with different excitonic coupling strength $|J|$ shows the anharmonicity of PES resulting from the excitonic coupling.

-I suggest to thoroughly revise the beginning of the section Results. Instead of starting from the Hamiltonian model description (that immediately avalanches specific jargon: Frenkel-Holstein, Condon's approximation, AM/PM, Fermi's Golden Rule, Huang-Rhys factors – by the way, the displacements S are not even introduced, etc.), the authors should provide an accessible discussion for a reader of the underpinning physical picture and outline advantages and limitations of the presented model.

Our response:

Thank you for your feedback and suggestion. We have thoroughly revised the beginning of the Results section. Firstly, we avoid using those unnecessary specific jargons. Secondly, we add more explanation on the physical meaning of our two-mode model used through this work. ***“One mode has a nonadiabatic coupling V , which couples the two electronic states of each monomer and induces the non-radiative decay. The other mode is coupled with the electronic states with electron-phonon coupling g .”*** Thirdly, we acknowledge that our two-mode model has limitations in neglecting low-frequency torsions that are crucial for flexible molecules. To address this, we conduct further simulations to validate our results under such circumstances. Fourthly, we also include discussions on the impact of inter-molecular motion, which was neglected in our model. This will provide a more comprehensive understanding of the system under study. In our revised manuscript, we aim to provide readers with a clearer understanding of the underlying physical picture related to the optimal excitonic coupling. Through systematic simulations and discussions, we not only explain the existence of this phenomenon, but also describe how its properties may vary under different conditions. Our hope is that this will enhance readers' comprehension of this

complex topic.

-Well, after reading further, the discussion about multiple vibrational modes does appear much later (page 13) and is partially addressed on the azulene example (again fused rings hydrocarbons), however, such arrangement negatively affects the readability of the paper

Our response:

Thank you for the helpful comment. We rearrange the manuscript structure, moving the azulene example to the front part of result section to discuss the validity of our simplified two-mode model and the extensibility of results from the simulation results before using this two-mode model in subsequent simulations of larger aggregate systems.

-Critique 2:

By continuing reading, I see

1. Eq. 1. H_1 is not introduced. One must read through the Method section to understand its meaning.

Our response:

Thank you for pointing out the mistake. we have moved detailed symbols and equations, such as H_1 , from the main text to the Methods section. While, in the Methods section, we provide a complete presentation of our method to the audience who are interested in them. The main text now avoids the use of such technical details and instead focuses on the conceptual understanding of our model and approach.

2. Eq (2) - the physical meaning of reorganization – is not explained at all. Vibrational creation/annihilation operators b are only introduced at the end, in the Methods section.

Our response:

Thank you for identifying the shortcomings. In the revised main text, we avoid technical details and instead focus on the conceptual understanding of vibrational distortion field.

“We characterize the effective reorganization energy of an aggregate based on the vibrational distortion field (VDF) $D_n(r)$, which is the magnitude of nuclear distortion of mode n at a distance r from an extra electron/exciton. VDF is well in accordance with the physical meaning of reorganization, which represents the change of equilibrium nuclear structure when the system transitions from one electronic state to the another. For the specific expression of $D_n(r)$, see Methods section. Then, the effective reorganization energy $\tilde{\lambda}$ of an aggregate is defined as $\tilde{\lambda} =$

$$\frac{1}{2} \sum_{n,r} \omega_n^2 D_n(r)^2.”$$

In the Methods section, the specific expression of VDF is given

$$D_n(r) = \left\langle \sum_i |S_1^i\rangle \langle S_1^i | \hat{q}_{i+r,n} \right\rangle_T. \quad (13)$$

“Here, i is the index of monomer, and $|S_1^i\rangle$ is the local electronic excited states of monomer i . $\hat{q}_{i+r,n}$ is the position operator of vibrational mode n in monomer $i+r$, which has a distance r from the excited monomer i . $\langle \cdot \rangle_T$ is the expectation value with respect to the thermal equilibrium initial state. Given the distortion $D_n(r)$, the effective reorganization energy of molecular aggregates is defined as $\tilde{\lambda} = \frac{1}{2} \sum_{n,r} \omega_n^2 D_n(r)^2$.”

3. Eq. (3) - vibrational frequencies are not introduced (the equation cannot be understood without Methods section).

Our response:

Thank you for pointing out our mistake. In the revised manuscript, we provide an explanation of the symbol ω as the vibrational frequency when it is first introduced. **“Our simulations utilize a simplified molecular aggregate model, in which each monomer is represented by a two-level electronic system coupled with two harmonic vibrational modes, unless otherwise specified. Both vibrational modes have a frequency of $\omega=1400\text{cm}^{-1}$, which is typical of C-C stretching vibrations and is regarded as the primary energy drain for non-radiative decay in organic dyes.”**

-Altogether the above 3 comments are suggestive that the authors started with quite technical manuscript and spitted a formal part into “Results” and “Methods”, which adversely impacted readability since one needs to jump from one place to another to understand the narrative.

To contrast, separation of the Main text and Methods should serve an opposite purpose: improved readability. The Main text should be self-contained (!) and written in accessible language, opening the paper to a broader audience such as experimentalists. In contrast the Methods should provide ‘in depth’ look to theory practitioners.

Bottom line: the text in Results section must be appropriately revised to address the above problems.

Our response:

Thank you for valuable instructions. We have thoroughly revised the manuscript structure and content based on your suggestions. In this revised manuscript, we have significantly improved the readability of the Results section by avoiding technical language and unnecessary symbols, and explaining the physical meaning of symbols when they are necessary to be included. We attempted to explain the phenomenon simply through its physical meaning instead of complex math equations. For example, a new schematic figure is presented to explain the same non-radiative behavior of J-aggregates and H-aggregates. Now our main text is self-contained and should be accessible to a broader audience. At the same time, we have provided more detailed technical information in the Methods section to provide an "in-depth" look to theory practitioners.

Please refer to the revised manuscript for more details.

-Critique 3:

Methods section has highly technical subsection B on the “Influence of the sign of excitonic coupling”, which is highly relevant to H- and J-aggregation. However, discussion of H- and J-

aggregate cases is largely missing from the Main text. This is an extremely important topic that needs to be addressed in accessible terms.

Our response:

Thank you for this important comment. In the revised manuscript, we discuss the selection rule of non-adiabatic coupling operator that causes the same non-radiative decay behavior in H-aggregate and J-aggregate with a schematic figure (Figure 3). We hope that these revisions will clarify the underlying physics and improve the readability of the manuscript.

First, we briefly explain the difference between H- and J-aggregates by *“The phase of the excitonic couplings between monomers depends on the arrangement of their transition dipole moments (TDM). For H-aggregates with sandwich-type TDM arrangement, J is positive, whereas for J-aggregates with head-to-tail TDM arrangement, J is negative. We examine the effects of both the magnitude and phase of excitonic couplings on the non-radiative decay of aggregates in our study.”*

Then we discuss why H- and J-aggregates display different radiative decay behavior but the same non-radiative behavior by comparing the coupling terms in two processes as

“Another noteworthy result is that J-aggregate ($J < 0$) and H-aggregate ($J > 0$) display the same non-radiative decay curve (see Supporting Information FIG.S1 for detailed comparisons). It is consistent with former studies. This differs from the widely known enhancement/inhibition behavior in the radiative decay of J-/H- aggregates, because the nonadiabatic coupling operator, unlike the dipole operator, has both electronic and nuclear parts, resulting in a selection rule distinct from that of radiative decay, as depicted in Figure 3. More specifically, because of excitonic coupling, the initial excited state is a linear combination of the local excited states. In the radiative decay process, one final state could have non-zero coupling with each component of the initial state (left panel in Figure 3), therefore the sign () of the linear combination determines whether the emission is enhanced (+) in J-aggregate or prohibited (-) in H-aggregate. On the contrary, in the non-radiative decay process, one final state only has non-zero coupling with one component of the initial state (right panel in Figure 3), making the sign of linear combination irrelevant. It should be noted that for non-radiative decay, the rigorous equivalence between J- and H- aggregates hold only when the modes with non-zero nonadiabatic coupling V all have zero electron-phonon couplings g . (An analytical derivation can be found in the Supporting Information section 1) Otherwise, J- and H- aggregates may display slightly different non-radiative decay behavior (see Supporting Information FIG.S1). Nevertheless, in real-world organic dyes with multiple vibrational modes, the modes with strong nonadiabatic couplings often have negligible electron-phonon couplings. Thus, we believe the sign of the excitonic coupling will have a minor effect on k_{nr}^{Agg} of real molecular aggregates. This is further supported by the following azulene example shown in Figure 4b.”

A schematic figure illustrating the different selection rule for radiative decay and non-radiative decay is also added as Figure 3 in the revised manuscript

Figure 3: A schematic figure illustrating the selection rule differences between the radiative decay and non-radiative decay processes of a dimer model.

Rectangles represent the monomers and the arrow inside indicates the orientation of the transition dipole moment. Each circle represents the vibrational mode with nonadiabatic coupling for each monomer. The red-colored rectangles and circles represent the local electronic and vibrational excited states, respectively. The grey arrows indicate coupling between the two states. The prohibition sign on the arrow means the coupling is zero.

The original Appendix B is moved to the Supporting information.

-Critique 4:

Toward the end of the Discussion section, it would be nice to connect formal theoretical findings on J_{OPT}/λ , etc. with realistic molecular aggregate systems for which most of the discussed quantities are known

Our response:

We appreciate this insightful suggestion. We connect our theoretical findings to a recent experiment [Michail et al., Phys. Chem. Chem. Phys. 22, 18340-18350 (2020)] and related discussions are added at the end of the Discussion section.

In this experiment, a series of squaraine J-aggregate homodimers (dSQA) with different bridge units are synthesized to control the distance and thus the excitonic coupling between the two monomers. The excitonic coupling strength $|J|$ and fluorescence quantum yield Φ_f have been experimentally measured. Here we calculated the monomer reorganization energy λ with density functional theory and time-dependent density functional theory to compare the ratio $|J|/\lambda$ and Φ_f across the series. Although the condensed phase effect and the distribution of vibration modes are neglected for the current simple analysis, we observe that the dSQA with the highest Φ_f had $|J|/\lambda \approx 0.5$, which is close to the optimal exciton coupling strength suggested by our theoretical studies. In the Supporting Information, Figure S11 illustrates the dSQA molecular structures and its photophysical properties in more detail. Of course, a detailed and rigorous study of this specific system is still necessary to further unveil the connection between experiments and theories.

Figure S11: On the left is the molecule structures of experiment synthesized dye SQA and its homodimer with various bridge units. On the right is the comparison on their photophysical properties.

REVIEWERS' COMMENTS

Reviewer #1 (Remarks to the Author):

The revised version is a better read and overall easier to grasp. The authors did a great job in modifying the text and adding additional results and graphics to clarify their results. My concerns are addressed but there are two new, minor points:

The additional part (b) in Fig. 1 does seem a bit removed from the rest of Fig. 1, in my opinion, as modes q_1 and q_2 only ever appear here and are not introduced. I am not sure if this brings across what the authors had in mind or rather leads to more confusion. I would remove it or shift it to the SI. But I leave this decision to the authors.

An additional minor thing is that the excitonic coupling J is not properly introduced in the "Model for Aggregates" section, i.e., the name and symbol is not properly connected.

Reviewer #3 (Remarks to the Author):

The authors honestly attempted to address all the Referee comments. As a result, the article has improved significantly, particularly, broad accessibility aspects. I nor recommend publication in Nature Comm. in its present form.

Response to Reviewers' Comments

Reviewer #1

The revised version is a better read and overall easier to grasp. The authors did a great job in modifying the text and adding additional results and graphics to clarify their results. My concerns are addressed but there are two new, minor points:

The additional part (b) in Fig. 1 does seem a bit removed from the rest of Fig. 1, in my opinion, as modes q_1 and q_2 only ever appear here and are not introduced. I am not sure if this brings across what the authors had in mind or rather leads to more confusion. I would remove it or shift it to the SI. But I leave this decision to the authors.

Our response:

We thank the Reviewer for the suggestion. We have additionally explained the meaning of “ q_1 ” and “ q_2 ” in the legend of FIG.1 in the revised manuscript as “A schematic diagram of the potential energy surface (PES) of the dimer model. In the left panel, the dashed curves represent the ground state PES, while the solid curves represent the two excited state PESs with displacement along mode q_1 and q_2 respectively compared with the ground state. q_1 / q_2 is the normal coordinate of two modes with electron-phonon coupling g of molecule 1 / molecule 2 (the vibration represented by the red circle in panel (a)) within the dimer model” We consider FIG. 1b shows the significant anharmonicity of aggregate excited state potential energy surface. This is one of the key difficulties for the evaluation of aggregate non-radiative decay rate and it can be solved by time-dependent density matrix renormalization group (TD-DMRG) used in this work. So we take it is necessary to appear in the main text.

An additional minor thing is that the excitonic coupling J is not properly introduced in the "Model for Aggregates" section, i.e., the name and symbol is not properly connected.

Our response:

Thank you for pointing out this mistake. In the revised manuscript, we have corrected it by connect the name and symbol at the first time they appear as “We only consider the excitonic coupling J between nearest-neighbor sites, since this coupling weakens with increasing inter-molecular distance.”

Reviewer #3

The authors honestly attempted to address all the Referee comments. As a result, the article has improved significantly, particularly, broad accessibility aspects. I nor recommend publication in Nature Comm. in its present form.

Our response:

Thank you for acknowledging our sincere efforts to address all the referee comments and improve the manuscript, particularly in terms of broad accessibility aspects. We appreciate your feedback.